# DIAGNOSING AND RECTIFYING VISION MODELS USING LANGUAGE

**Yuhui Zhang, Jeff Z. HaoChen, Shih-Cheng Huang, Kuan-Chieh Wang, James Zou, Serena Yeung**
Stanford University, Stanford, CA 94305, USA
`{yuhuiz, jhaochen, mschuang, wangkua1, jamesz, syyeung}@stanford.edu`

## ABSTRACT

Recent multi-modal contrastive learning models have demonstrated the ability to learn an embedding space suitable for building strong vision classifiers, by leveraging the rich information in large-scale image-caption datasets. Our work highlights a distinct advantage of this multi-modal embedding space: the ability to diagnose vision classifiers through natural language. The traditional process of diagnosing model behaviors in deployment settings involves labor-intensive data acquisition and annotation. Our proposed method can discover high-error data slices, identify influential attributes and further rectify undesirable model behaviors, without requiring any visual data. Through a combination of theoretical explanation and empirical verification, we present conditions under which classifiers trained on embeddings from one modality can be equivalently applied to embeddings from another modality. On a range of image datasets with known error slices, we demonstrate that our method can effectively identify the error slices and influential attributes, and can further use language to rectify failure modes of the classifier.

## 1 INTRODUCTION

Recent models trained using multi-modal contrastive learning have leveraged large-scale datasets of aligned image-caption pairs to obtain shared embedding spaces that capture rich visual and textual features. The learned image and text encoders resulting from multi-modal contrastive learning have been demonstrated to be effective feature extractors that can be used to train strong single-modality classifiers (Radford et al., 2021; Jia et al., 2021; Yuan et al., 2021). In this work, we show how visual classification models obtained through multi-modal contrastive learning, as described above, offer a significant additional advantage: the ability to use language to probe and diagnose the behavior of the vision models.

Model diagnosis aims to gain a systematic and comprehensive understanding of when and why models fail. This is a critical quality assurance process to prevent unexpected and catastrophic failures of models in high-stake settings. A growing body of work has proposed methods for addressing this need. For example, error slice discovery methods aim to find subsets of inputs with similar characteristics where the model performs significantly worse (d'Eon et al., 2022; Eyuboglu et al., 2022). Interpretability methods aim to understand the black-box process of model prediction and thus the reasons why models fail for certain inputs (Ribeiro et al., 2016; Lundberg & Lee, 2017; Koh et al., 2020). In addition, model diagnosis is relevant to *model auditing*, an important topic that also deals with identifying model failures and sensitive attributes (Raji et al., 2020), and has a broad societal impact in terms of AI accountability and integration (Buolamwini & Gebru, 2018; Mitchell et al., 2019; Gebru et al., 2021).

While these prior efforts have made progress in vision model diagnosis, they all suffer from a critical Achilles' heel — *susceptibility to lack of visual data*. Curated training and test sets from the same data distribution are typically used to develop vision models. Even if models achieve perfect performance on these datasets, their performance can degrade drastically when deployed in-the-wild, due to distribution shifts (Koh et al., 2021; Wiles et al., 2022). Yet most existing model diagnosis methods require visual examples of failure modes (e.g., present in the test set) to discover them. As

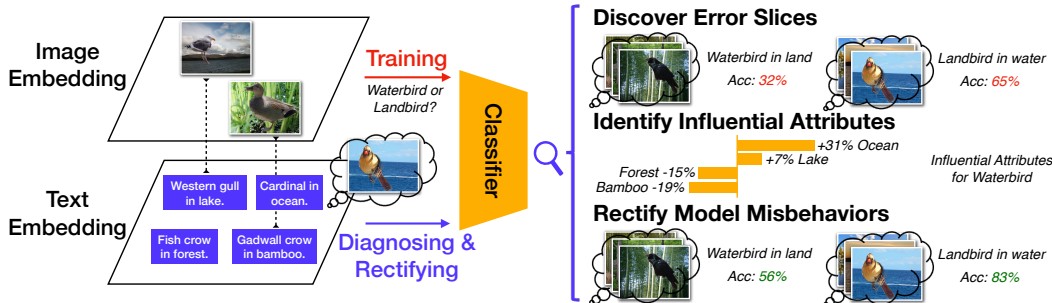

Figure 1: **Overview of our approach, DrML, that diagnoses and rectifies vision models using language.** Our approach leverages the shared image and text representation space learned by multi-modal contrastive learning. We find that classifiers trained on embeddings from one modality can be equivalently applied to embeddings from another modality, despite the fact that embeddings from these two modalities are distinctly separated. This cross-modal transferability phenomenon enables us to diagnose a vision model by training it on the image embedding space and probing it with text embeddings. The use of language allows us to generate a large set of diverse and novel inputs to discover error slices, identify influential attributes, and rectify model misbehaviors.

a result, using these methods is reliant on efforts to collect large-enough datasets to cover all data distributions and potential failure modes of interest, which is often impractical or infeasible.

The goal of our work is to circumvent this need to collect test data representing all data distributions of interest, and instead use natural language input to diagnose vision classifiers. It is often easier to generate a set of diverse natural language inputs by combining known attributes and prompt generators than to collect a set of image inputs representing the same desired concepts. We observe that vision classifiers trained on image embeddings from a shared image-text embedding space suggest the possibility of leveraging text embeddings as a proxy for image embeddings. Multi-modal contrastive losses are frequently used to learn such shared embedding spaces. However, while these losses encourage image and text embeddings to be closer for aligned pairs than for mismatched pairs, there is no guarantee that in practice, using text embeddings as input into a vision classifier trained on the image embeddings will result in the same predictions. In this work, we first verify that text inputs can indeed work as good proxies to image inputs trained on a shared image-text embedding space obtained through contrastive learning. We refer to this as *cross-modal transferability*.

Based on the phenomenon of cross-modal transferability, we then present **DrML** for **D**iagnosing and **R**ectifying Vision **M**odels using **L**anguage. We show that DrML can use language to *diagnose* vision models in two different ways: discovering error slices including concepts for which we have no visual data, and identifying attributes that have the greatest impact on model predictions. Finally, we present a method that uses language to *rectify* undesirable behaviors without requiring the collection of more visual data. Figure 1 illustrates our framework for diagnosing and rectifying vision models using language. On three image datasets representing the three most common types of model failure modes, we demonstrate that DrML can effectively identify error slices and influential attributes, and can further rectify these model failure modes using language.

In summary, our contributions are:

1. We present a theoretical explanation of when cross-modal transferability happens (Section 2.1), and empirically verify that the assumptions required by the analysis is true in practice across a range of multi-modal contrastive models and datasets (Section 3.2).

2. We propose DrML, a framework for diagnosing vision models using natural language, including error slice discovery and influential attribute identification. We empirically validate DrML by simulating common types of failure modes using the Waterbirds (Sagawa et al., 2020), Fair-Face (Karkkainen & Joo, 2021), and dSpitesV (Matthey et al., 2017) datasets, and show the effectiveness of our method in identifying known error slices and influential attributes.

3. We further demonstrate that DrML can rectify undesirable model behaviors and improve model performance with respect to the identified error slices and influential attributes, by fine-tuning the vision classifier using text embeddings constructed from the diagnosis process.

## 2 APPROACH

We first define basic notations used in this paper. Given a pre-trained multi-modal contrastive model, along with an image $X \in \mathcal{X}$ or text $Y \in \mathcal{Y}$ as input, we can obtain their $l_2$-normalized embeddings $\boldsymbol{x}$ or $\boldsymbol{y}$ from the image encoder $f_x : \mathcal{X} \mapsto \mathbb{R}^d$ or the text encoder $f_y : \mathcal{Y} \mapsto \mathbb{R}^d$, respectively, where $d$ is the dimension of the shared multi-modal embedding space. We can build classifiers $h : \mathbb{R}^d \mapsto \mathcal{C}$ such as a linear layer or multi-layer perception on the shared embedding space to predict the label $c \in \mathcal{C}$ given an image embedding or text embedding. We focus on the case of vision classifiers trained using image embeddings.

### 2.1 TEXT EMBEDDINGS AS PROXIES FOR IMAGE EMBEDDINGS

The core of our work hinges on the ability to use text as a proxy for image inputs, thereby enabling us to use language to diagnose vision models. Here we describe our approach to analyze if this is feasible in practice — *are text inputs good proxies for images in contrastive representation space?*

**Cross-modal Transferability.** To answer the question, we first define cross-modal transferability. Let $P_{\mathcal{D}}$ be the joint data distribution over image-text pairs. For $X, Y \sim P_{\mathcal{D}}$, we denote $\boldsymbol{x} = f_x(X)$ and $\boldsymbol{y} = f_y(Y)$ the corresponding image and text embeddings respectively. We say that a vision classifier $h$ achieves cross-modal transferability when it outputs similar predictions on $\boldsymbol{x}$ and $\boldsymbol{y}$. In other words, the difference across the prediction pair is small:

$$\mathbb{E}_{\boldsymbol{x}, \boldsymbol{y}}[D(h(\boldsymbol{x}), h(\boldsymbol{y}))] \approx 0,$$

where $D(\cdot, \cdot)$ measures the difference between predictions, e.g. the 0-1 loss $D(u, v) = \mathbf{1}_{u \neq v}$.

**Modality Gap.** While intuition suggests that embeddings of a matched image-caption pair should be close, recent work shows instead that the embeddings are approximately clustered per modality Liang et al. (2022). They refer to the distance between these clusters as the *modality gap*. We define the individual-level modality gap $\boldsymbol{g}$ as the difference between image and text embeddings for a single pair, and the class-level gap $\boldsymbol{g}_c$ as the average difference between image and text embeddings for a given class $c \in \mathcal{C}$. Formally, the modality gap definitions are written as:

$$\boldsymbol{g} = \boldsymbol{x} - \boldsymbol{y} \text{ and } \boldsymbol{g}_c = \boldsymbol{x}_c - \boldsymbol{y}_c, \text{ where}$$
$$\boldsymbol{x}_c = \mathbb{E}_{X \sim P_{\mathcal{D}}(X|c)}[f_x(X)], \quad \boldsymbol{y}_c = \mathbb{E}_{Y \sim P_{\mathcal{D}}(Y|c)}[f_y(Y)].$$

**Modality Gap Geometry.** We take a closer look at the modality gap geometry across a range of multi-modal contrastive models and datasets, presented in detail in Section 3.2, and empirically find that the following hold true:

1. *The modality gap between corresponding image and text embeddings can be approximated by a constant vector, particularly at the class level.* We verify this by computing distributions over $\|\boldsymbol{g}\|$ (*magnitude*) and $\cos(\boldsymbol{g}, \mathbb{E}_{\boldsymbol{g}}[\boldsymbol{g}])$ (*direction*).

2. *The modality gap is orthogonal to the span of image embeddings and text embeddings, and image embeddings and text embeddings have zero mean in the subspace orthogonal to the modality gap.* We verify this by computing distributions over $\cos(\boldsymbol{x} - \mathbb{E}_{\boldsymbol{x}}[\boldsymbol{x}], \mathbb{E}_{\boldsymbol{g}}[\boldsymbol{g}])$ (*orthogonality*) and $\mathbb{E}_{\boldsymbol{x}}[\boldsymbol{x} - \boldsymbol{x}^T \boldsymbol{g}' \boldsymbol{g}']_i$ (*center*), where $\boldsymbol{g}' = \mathbb{E}_{\boldsymbol{g}}[\boldsymbol{g}]/\|\mathbb{E}_{\boldsymbol{g}}[\boldsymbol{g}]\|$ and $i \in [d]$. The subscript $i$ denotes indexing the $i$-th dimension of the vector.

**Cross-modal Transferability under Modality Gap.** The above findings with respect to the geometry of the modality gap indicate that the classifier input between training and cross-modal evaluation only differs in a constant $\boldsymbol{g}$, i.e., $h(\boldsymbol{x}) \approx h(\boldsymbol{y} + \boldsymbol{g})$. Intuitively, since the modality gap $\boldsymbol{g}$ is an orthogonal constant to the span of embeddings, the weight matrix of the learned classifier should also be orthogonal to $\boldsymbol{g}$. Hence the prediction of the classifier is not affected by $\boldsymbol{g}$. This intuition explains why we observe strong cross-modal transferability under modality gap in practice, across different multi-modal contrastive models trained on different datasets. These results are presented in Section 3.2. In the following Proposition 2.1, we further theoretically prove that a linear classifier trained with a regularized quadratic loss is *guaranteed* to be orthogonal to the modality gap and hence achieves cross-modal transferability. The formal statement and proof are in Appendix A.2.

**Proposition 2.1** (Informal version of Proposition A.1)**.** *Suppose there exists a gap vector $\boldsymbol{g} \in \mathbb{R}^d$ such that every pair of image embedding $\boldsymbol{x}$ and caption embedding $\boldsymbol{y}$ satisfies $\boldsymbol{g} = \boldsymbol{x} - \boldsymbol{y}$, the gap*

*$g$ is orthogonal to the span of image embeddings, and the image embeddings have zero mean in the subspace orthogonal to $g$. Then, any linear function minimizing a regularized quadratic loss on image embeddings achieves the same loss on text embeddings, enabling cross-modal transferability.*

**Cross-modal Transferability by Closing the Modality Gap.** The observation that the modality gap approximates a constant provides us another perspective to achieve cross-modal transferability — by closing the modality gap so that there is no inconsistency when feeding embeddings from another modality. We propose a simple technique to close the modality gap so that the gap becomes zero. During training, instead of feeding $x$ to the model $h$, we feed it with $x - \mathbb{E}_x[x]$. During cross-modal evaluation, we feed $y - \mathbb{E}_y[y]$ instead of $y$. With this strategy, we close the gap and observe additional improvements in cross-modal transferability compared to training with the gap.

## 2.2 DIAGNOSING VISION MODELS USING LANGUAGE

Having established that text embeddings can be good proxies for image embeddings (Section 2.1 and 3.2), we now describe DrML, which uses natural language inputs for diagnosing vision classifers.

**Discovering Error Slices through Language.** Deep learning models often make systematic errors on subgroups of inputs with similar attributes, referred to as *error slices* and formally defined as:

$$\mathbb{S} = \{\mathcal{S} \subseteq \mathcal{X} | e(\mathcal{S}) \gg e(\mathcal{X})\},$$

where $\mathcal{X}$ is a test set of images and $e(\cdot)$ is the model's error rate on the set of input images. However, collecting a large enough test set that covers different image distributions is a fundamental challenge. The collected test set often only covers a small percentage of model failure modes (i.e., error slices) in the wild. In contrast, language inputs are easy to generate.

Our proposed method, DrML, is capable of discovering error slices through language inputs. DrML works as follows:

1. We define an *attribute set* $\mathcal{A}$ related to the task.

2. Given a specific attribute subset $\mathcal{F} \subseteq \mathcal{A}$, we use different prompt generators $p \in \mathcal{P} : 2^{\mathcal{A}} \mapsto \mathcal{Y}$ to map attribute combinations to text inputs.

In this way, we can combine a wide range of attributes with different prompts to collect a diverse and novel set of text inputs. The generated text set $\mathcal{Y}$ is typically much more diverse than the available image test set $\mathcal{X}$, allowing the discovery of more comprehensive and unseen error slices.

Importantly, DrML has two distinctive benefits over the typical approach of using an image test set. First, DrML only requires minimal effort to define a meaningful set of attributes to generate the input set, circumventing the human cost of data collection. Second, the combination of defined attributes naturally defines human-interpretable data slices, whereas image-based slice discovery methods do not directly provide a text summary of the error slice.

**Identifying Influential Attributes through Language.** Interpreting what attributes influence model predictions is crucial for understanding why models fail. Since language is directly interpretable by humans, we perform counterfactual analysis using language to understand which attributes or concepts most impact model predictions. With $\mathcal{A}$ defined as the attribute set, we aim to identify a subset of attributes that significantly influences model predictions to any given class $c$:

$$\mathbb{A}_c = \{a \in \mathcal{A} | s_c(a) \gg 0\},$$

where $s_c(\cdot)$ is the *influence* of an attribute to class $c$. We measure the influence by Shapley value, a widely-used interpretation tool in machine learning (Lundberg & Lee, 2017; Ghorbani & Zou, 2019), which computes average prediction change with the presence and absence of this attribute:

$$s_c(a) = \sum_{\mathcal{F} \subseteq \mathcal{A} \setminus \{a\}} \frac{|\mathcal{F}|!(|\mathcal{A}| - |\mathcal{F}| - 1)!}{|\mathcal{A}|!} (p_c(\mathcal{F} \cup \{a\}) - p_c(\mathcal{F})),$$

where $p_c(\cdot)$ is the average predicted probability of class $c$ on a set of inputs with certain attributes.

With natural language, we can easily compose a large set of inputs with and without that attribute and feed them to the model to calculate the influence. For example, to compute the influence of attribute

"ocean" on class "waterbird", we can generate various text inputs such as "A photo of *species* on the ocean" and "A photo of *species*", and compute the average difference of the model predicted probabilities of "waterbird". Note that it is particularly challenging to identify influential attributes using image inputs because it requires an extensive collection of images with attribute annotations.

**Connection.** Discovering error slices and identifying influential attributes are important complementary applications, with the same ultimate goal —— diagnosing the model. Error slice discovery finds specific subgroups about when the model fails, while attributes provide abstract explanations of why the model fails. Meanwhile, figuring out influential attributes helps discover error slices, because attributes provide information about the space of potential error slices, and vice versa.

## 2.3 Rectifying Vision Models using Language

In addition to discovering errors during model diagnosis, how to rectify these errors is a practical but challenging problem, which is seldomly addressed in existing works about the model diagnosis. Our finding about cross-modal transferability enables us to rectify undesirable behaviors of vision classifiers through language. Here we propose a simple solution where we generate additional data that the model fails using language and continue training the model on these synthesized data.

Given the error slices $\mathbb{S} = \{\mathcal{S} \subseteq \mathcal{X} | e(\mathcal{S}) \gg e(\mathcal{X})\}$ discovered, we aim to rectify model performance on these error slices by minimizing $|\mathbb{S}|$. For each $\mathcal{S} \in \mathbb{S}$ defined by a list of attributes, we generate a large set of natural language inputs related to this slice $\mathcal{Y}_\mathcal{S}$ through attribute composition and prompt manipulation (Appendix B) and continue training the model on these text inputs $\mathcal{Y}_\mathcal{S}$. We continue training the model using the same hyperparameters as if the model is trained on corresponding images, since we have proved that texts are effective proxies of images. This simple strategy significantly improves model performances on corresponding image error slices with minimal impact on other data, and has a distinct advantage that no visual data is required for rectification.

## 3 Experiments

In this section, we first demonstrate that text embeddings are good proxies for image embeddings in multi-modal contrastive representation space (Section 3.2). Based on that, we demonstrate how DrML successfully discovers error slices (Section 3.3), identifies influential attributes (Section 3.4), and further rectifies model misbehaviors on three datasets (Section 3.5).

### 3.1 Experimental Details

**Model Architecture.** We use CLIP (Radford et al., 2021) as the shared multi-modal embedding space. For classifiers built on CLIP's embeddings, we use linear layers and multi-layer perceptrons.

**Datasets.** For cross-modality transferability (Section 3.2), we use the **MS-COCO** dataset (Lin et al., 2014), which includes both captions and object annotations for each image. The task is a multi-label classification problem of predicting the presence of 80 objects based on images or captions. For model diagnosis and rectification, we simulate the three common types of model failures. For *spurious correlation*, we use the **Waterbirds** dataset (Sagawa et al., 2020) which asks a model to classify if a given bird image is a waterbird or a landbird. The training data contains a spurious correlation between bird species and backgrounds — 95% of waterbirds appear in the water, and 95% of landbirds appear on the land. For *underrepresented data*, we use **FairFaces** (Karkkainen & Joo, 2021) which contains face images from 9 age groups and 7 race groups. The task is gender classification. To simulate the underrepresentation of minority groups, we sample races in proportion to the demographics of the state of Montana for our training set. For *unseen data*, we use **dSpritesV** (Matthey et al., 2017) which contains images of shapes with different colors, sizes, and positions. The task is to classify the shape in an image. To simulate errors caused by unseen data, we only use images with orange triangles or green squares during training. More details are shown in the Appendix B.

### 3.2 Are Text Embeddings Good Proxies for Images Embeddings?

We have provided theoretical explanations in Section 2.1 that a classifier's boundary is transferable across modalities if the modality gap satisfies certain geometric conditions. Here we first verify these conditions and then show empirically that closing the modality gap can improve transferability.

| Model | Magnitude | | Direction | | Orthog- | Center |
|---|---|---|---|---|---|---|
| | Individual | Class | Individual | Class | onality | |
| CLIP COCO (2021) | $1.18 \pm 0.03$ | $0.88 \pm 0.04$ | $0.70 \pm 0.06$ | $0.94 \pm 0.04$ | $0.00 \pm 0.06$ | $0.00 \pm 0.02$ |
| CLIP ImageNet (2021) | - | $1.00 \pm 0.02$ | - | $0.83 \pm 0.05$ | $0.00 \pm 0.06$ | $0.00 \pm 0.03$ |
| ConVIRT (2022) | $1.22 \pm 0.10$ | - | $0.67 \pm 0.09$ | - | $0.02 \pm 0.10$ | $0.00 \pm 0.02$ |
| VideoCLIP (2021) | $1.35 \pm 0.03$ | - | $0.79 \pm 0.04$ | - | $0.00 \pm 0.06$ | $0.00 \pm 0.02$ |
| CLASP (2021) | $1.33 \pm 0.04$ | - | $0.79 \pm 0.12$ | - | $0.03 \pm 0.13$ | $0.00 \pm 0.02$ |

Table 1: **Geometry analysis of modality gap for various multi-modal contrastive representation spaces.** The modality gap approximates a constant vector, indicated by the magnitude and direction distributions. Modality gap is also orthogonal to the span of embeddings from two modalities, and embeddings' centers for both two modalities are zero vectors in the subspace orthogonal to the gap, indicated by the orthogonality and center distributions. Based on our theoretical analysis, these findings suggest that cross-modal transferability is widely established in multi-modal contrastive learning. $\pm$ connects mean and standard deviation. Detailed distributions in Figure 3.

| Modality Gap | Model | Evaluation on Image | | | Evaluation on Text | | | Consistency$_\uparrow$ |
|---|---|---|---|---|---|---|---|---|
| | | Loss$_\downarrow$ | mF1$_\uparrow$ | MF1$_\uparrow$ | Loss$_\downarrow$ | mF1$_\uparrow$ | MF1$_\uparrow$ | |
| - | Random | 0.6939 | 0.0655 | 0.0443 | 0.6938 | 0.0696 | 0.0437 | 0.8644 |
| Default | Linear | 0.0501 | 0.7276 | 0.6790 | 0.1188 | 0.5642 | 0.5429 | 0.9637 |
| | MLP | 0.0480 | 0.7523 | 0.7158 | 0.0888 | 0.6350 | 0.6135 | 0.9789 |
| Closing | Linear | 0.0498 | 0.7280 | 0.6777 | **0.0719** | **0.6554** | **0.6168** | **0.9842** |
| | MLP | 0.0483 | 0.7495 | 0.7130 | **0.0885** | **0.6503** | **0.6358** | **0.9806** |

Table 2: **Cross-modal transferability in multi-modal contrastive representation learning.** We train a classifier using CLIP's image embeddings and test the trained classifier using text embeddings on the MS-COCO multi-label classification dataset. Despite the modality gap, classification boundaries learned from one modality are transferable to another modality. Closing the modality gap further improves cross-modal transferability without harm to in-modal evaluation. Notations: mF1 - Micro F1, MF1 - Macro F1, Random - A randomly initialized linear classifier.

**Modality Gap Geometry.** In Table 1, we first show that *the modality gap can be well approximately by a constant vector*. For instance, on MS-COCO, the class-level gaps between image and text embeddings extracted from CLIP (ViT-B/32) have almost the same magnitude ($0.88 \pm 0.04$) and direction (cosine similarity $0.94 \pm 0.04$). We then show that *the modality gap is orthogonal to the span of image embeddings and text embeddings, and embeddings have zero mean in the subspace orthogonal to modality gap*. This is supported by the near-zero means with low standard deviations in "orthogonality" and "center" columns. Our findings here show that the assumptions required by our theory of cross-modal transferability (Section 2.1) hold true in practice across various datasets and contrastive multi-modal models, suggesting that *cross-modal transferability should be a pervasive phenomenon in multi-modal contrastive learning*.

**Cross-modal Transferability.** Table 2 shows the image-to-text transfer results on the MS-COCO validation set. Based on our theory, we indeed find that *cross-modality transferability is possible regardless of the modality gap*. For instance, we find that an image-embeddings-trained linear classifier capable of achieving 67.90% macro F1 score can maintain 54.29% macro F1 score using text embeddings as inputs, and the consistency between predictions using images and texts is 96.37%. Similarly, text-to-image transfer is also possible, which is shown in Appendix Table 7. While there exists slight degradation in performance under cross-modal evaluation, the difference in performance is relatively small, and the cross-modal transfer performance is much higher than random classification. The same finding is observed when using multi-layer perceptrons that learn non-linear features. As shown in the bottom half of Table 2, closing the modality gap further improves cross-modal transferability. The linear classifier achieves 9.12%, 7.39%, and 2.05% absolute improvements on micro F1, macro F1, and prediction consistency for image-to-text transfer without harm to in-modality evaluation. The improvements using MLP are smaller but consistent.

**Are Generated Language Prompts Good Predictors of Error Slices?** Here we further investigate whether our generated language prompts are good predictors of the error rate of a given data slice. We do so by looking at the correlation between performances on *generated* prompts and corresponding image slices. A strong correlation indicates that we can perform error slice discovery using text as proxies, which circumvents the challenges of collecting image data.

| Method | Waterbirds | | FairFace | | dSpritesV | |
|---|---|---|---|---|---|---|
| | Spearman | Pearson | Spearman | Pearson | Spearman | Pearson |
| (Base): Gen 1 Image (Prob) | 0.5822 | 0.5608 | 0.3884 | 0.3361 | 0.3059 | 0.3119 |
| (Base): Gen 20 Images (Prob) | 0.6034 | 0.5938 | 0.4288 | 0.5411 | 0.4557 | 0.5309 |
| (1): Generate 1 Text for Slice | 0.4167 | 0.4355 | 0.0801 | 0.0957 | 0.6278 | 0.6723 |
| (2): (1) + Use Label Probability | 0.5899 | 0.5773 | 0.2065 | 0.1760 | **0.7071** | 0.7481 |
| (3): (2) + Prompt Engineering | **0.6462** | **0.6721** | **0.5669** | 0.7024 | 0.6998 | 0.7595 |
| (Ours): (3) + Prompt Ensemble | **0.6465** | **0.6776** | 0.5614 | **0.7227** | **0.7028** | **0.7918** |

Table 3: **Correlation analysis of model performance on image and text slices.** Correlation can be improved by using label probability instead of label accuracy on text predictions, generating better text through prompt engineering and ensemble. Our approach outperforms the baseline text-to-image generation model by a large margin. The best or near-best results are bolded.

We treat each attribute subset $\mathcal{F} \subseteq \mathcal{A}$ as a slice. For each slice, we generate a set of text inputs $\mathcal{Y}_{\mathcal{F}}$ using prompt generators $\mathcal{P}$ and select all the images $\mathcal{X}_{\mathcal{F}}$ with attributes $\mathcal{F}$. We compute the Spearman and Pearson correlation between model performances on $\mathcal{Y}_{\mathcal{F}}$ and $\mathcal{X}_{\mathcal{F}}$. Table 3 shows strong correlation between image and text slices. Furthermore, correlation can be improved by: 1) using the average probability of the label on text predictions instead of accuracy, 2) generating better text inputs via prompt engineering which composes attributes into a more fluent sentence, and 3) prompt ensemble that uses different prompts to generate more diverse inputs (details in Appendix B).

As baselines for comparison, we use the state-of-the-art text-to-image generation model (Rombach et al., 2022) $t : \mathcal{Y} \mapsto \mathcal{X}$ to generate a set of (we use 1 or 20 in our experiment) images $\mathcal{X}'_{\mathcal{F}}$ from text prompts $\mathcal{Y}_{\mathcal{F}}$ and compute correlations between $\mathcal{X}'_{\mathcal{F}}$ and $\mathcal{X}_{\mathcal{F}}$. Our method outperforms this baseline by a large margin and does not utilize significant computational time and cost typically required for the image generation process. Samples of the generated image samples are shown in Appendix C. Even while significant progress has been made in text-to-image generation, generating high-fidelity images that maintain the original semantics is still challenging.

In summary, combining the empirical findings presented in this section and the theoretical results in Section 2.1, we show that text inputs can act as good proxies for image inputs, enabling us to diagnose vision classifiers using generated language prompts.

## 3.3 DISCOVERED ERROR SLICES

The strong correlation between the performances on text and image slices allows us to confidently run image slice discovery algorithm using text inputs. In this study, we use a simple error slice discovery method of sorting slices by their performances. We further marginalize attributes by merging similar slices into larger slices. In Table 4, we summarize the most essential discovered error slices by our language-based approach on the three datasets, each representing one of the three typical model failure patterns under distribution shifts (Wiles et al., 2022).

For **Waterbird**, the top identified error slices are waterbirds in land and landbirds in water, which correctly corresponded to errors are caused by *spurious correlations* present in the dataset. For **FairFace**, the African American population is among the top identified error slice, which also reflects their *underrepresentation* in our training set. For **dSpitesV**, our method correctly identifies green triangles and orange square as critical error slices. Additionally, pink triangle slices are also correctly identified, since they were *never seen* in the training data. By using images to verify our discovered slices, our method not only correctly identifies the most critical error slices but also accurately predicts the slice performances on images.

In Appendix C, we report results from the state-of-the-art slice discovery baseline DOMINO (Eyuboglu et al., 2022). When evaluated using datasets with the same distribution as the training set, DOMINO can only discover slices present in the dataset, and could not discovery errors caused by distribution shifts.

## 3.4 IDENTIFIED INFLUENTIAL ATTRIBUTES

In Table 5, we report the most influential attributes to a specific class on the same three datasets. These attributes provide a high-quality interpretation of how models predict and why they fail. For

| | Waterbirds | | | FairFace | | | dSpritesV | | |
|---|---|---|---|---|---|---|---|---|---|
| Slice | Text | Image | Slice | Text | Image | Slice | Text | Image |
| **Wbird in L** | **0.3258** | **0.3233** | **Black** | **0.9134** | **0.8997** | **Green triangle** | **0.1641** | **0.0616** |
| **Lbird in W** | **0.7029** | **0.6524** | Indian | 0.9268 | 0.9446 | **Orange square** | **0.2990** | **0.0337** |
| Wbird in W | 0.9306 | 0.9549 | Asian | 0.9305 | 0.9381 | **Pink triangle** | **0.5044** | **0.9861** |
| Lbird in L | 0.9957 | 0.9979 | White | 0.9427 | 0.9597 | Red triangle | 0.5651 | 0.9954 |

Table 4: **Discovered error slices using language.** With the images used for validation, our method succeeds in discovering important error slices (bolded) and accurately predicts model performances on image slices. Notations: Image - model accuracy using real image inputs, Text-predicted accuracy using text inputs as a proxy, W - water, L - land.

| | Waterbirds (waterbird) | | | FairFace (female) | | | dSpritesV (triangle) | |
|---|---|---|---|---|---|---|---|---|
| | Attribute | Influence | | Attribute | Influence | | Attribute | Influence |
| **Place** | Ocean | 0.3062 | **Age** | Very old | 0.0229 | **Color** | Orange | 0.3736 |
| | Lake natural | 0.0713 | | Young | 0.0161 | | Red | -0.0470 |
| | Forest broadleaf | -0.1540 | | Little | -0.0079 | | Pink | -0.1181 |
| | Bamboo forest | -0.1931 | | Infant | -0.0171 | | Green | -0.3321 |

Table 5: **Identified influential attributes using language.** We show top 2 most positively and negatively influential attributes, which provide insights into how models predict and why they fail.

| | Waterbirds | | | | | FairFace | | | | |
|---|---|---|---|---|---|---|---|---|---|---|
| Slice | Original | Rectify | Lonly | JTT | GDRO | Slice | Original | Rectify | Lonly | JTT | GDRO |
| **Wbird in L** | 0.3233 | 0.5564 | 0.5639 | **0.5865** | 0.7368 | **Black** | 0.8997 | **0.9075** | 0.8920 | 0.8920 | 0.9017 |
| **Lbird in W** | 0.6524 | **0.8271** | 0.7747 | 0.7468 | 0.8155 | Asian | 0.9381 | **0.9390** | 0.9290 | **0.9390** | 0.9384 |
| Wbird in W | 0.9549 | **0.9700** | 0.9023 | 0.9248 | 0.9474 | Indian | **0.9446** | 0.9439 | 0.9301 | 0.9406 | 0.9453 |
| Lbird in L | **0.9979** | 0.9893 | 0.9443 | 0.9764 | 0.9443 | White | 0.9597 | **0.9626** | 0.9424 | 0.9602 | 0.9588 |

Table 6: **Rectified model performances on discovered error slices.** We continue training models on language inputs corresponding to error slices (bolded) and observe significant performance improvements on these slices. GDRO not directly comparable because attribute annotations required.

example, one of the most influential attributes for waterbird classification is "ocean" with an influence value of 0.3062, indicating that the *model predicted probability of waterbird increases by 0.3* on average when "ocean" is present in a bird image. Since the attribute "place" should not affect predictions, this shows an obvious error of the model. Similar findings apply to the attribute "color" for dSpitesV. But what is more interesting is that the color "pink" is *never seen* during training but will bias the model to predict "square" with 0.1 increased probability. On FairFace, no attribute is found to significantly influence model prediction; thus, no obvious spurious correlations were learned.

### 3.5 RECTIFIED MODEL MISBEHAVIORS

In Table 6, we report performances of original models and rectified models. On both Waterbirds and FairFace dataset, our simple method of continue training the model on text inputs significantly improves model performances on error slices with minor influences on other slices. We also perform ablation by only training the model on all the language inputs from scratch (Lonly), and find that continuing to train the pre-trained image model achieves better results, but even training only with language can also work reasonably.

Our approach rectifies model misbehaviors caused by spurious correlation and underrepresented data by correcting the data bias. Another series of methods to tackle these errors are robust training techniques, such as GDRO (Sagawa et al., 2020) and JTT (Liu et al., 2021), which explicitly optimizes each slice's performance during training. Our method outperforms JTT. While GDRO performs similarly to ours, it requires attribute annotations on images, which is highly time-consuming and cost-prohibitive for most real-world applications. Moreover, GDRO and JTT cannot fix errors on unseen data, while ours can, because our rectifying process requires no visual data.

## 4 RELATED WORK & DISCUSSION

**Multi-modal Contrastive Learning.** Many recent works in vision-language contrastive learning, such as CLIP (Radford et al., 2021), ALIGN (Jia et al., 2021), and Florence (Yuan et al., 2021),

have leveraged large image-caption datasets to obtain embedding spaces that capture rich visual and textual features. As a result, the learned image and text encoders are demonstrated to be strong uni-modal classifiers. In this work, we show how vision models obtained through multi-modal contrastive learning offer another significant advantage — model diagnosis and rectification.

**Multi-modal Contrastive Representation Space Geometry.** Although multi-modal contrastive learning minimizes the distance between embeddings for matched pairs, prior work has shown that embeddings from two modalities are distinctively separated in the embedding space, which is referred to as modality gap (Liang et al., 2022). In this work, we further analyze the modality gap geometry and connect it to the cross-modal transferability phenomenon. Our finding is related to several recent works built on multi-modal contrastive representation spaces, such as DALL-E 2 (Ramesh et al., 2022), ClipCap (Mokady et al., 2021), and other works (Cohen et al., 2022; Gal et al., 2022). They found that trained models can directly take cross-modal embeddings but worse than same-modal embeddings. We not only explain this but provide a straightforward solution to improve transferability, which can be applied to all future works built upon multi-modal embeddings.

**Slice Discovery.** Many recent works aim to understand model systematic errors by finding subsets of inputs with similar characteristics where the model performs significantly worse. This is referred to as slice discovery (Chung et al., 2019; Singla et al., 2021; d'Eon et al., 2022; Eyuboglu et al., 2022; Jain et al., 2022a). However, these algorithms fail to address the most fundamental challenge for slice discovery — the lack of data. These works are only able to find errors that exist in the dataset. Our work circumvents the data challenge by performing slice discovery on the text space.

**Interpretation.** Many model interpretation methods have been proposed, including attribution-based (Ribeiro et al., 2016; Lundberg & Lee, 2017; Shrikumar et al., 2017) and concept-based (Ghorbani et al., 2019b; Koh et al., 2020). While these methods help in understanding the model prediction process, the outputs are complicated for humans to understand and inconsistent across models and algorithms (Ghorbani et al., 2019a; Jain et al., 2022b; Joshi et al., 2021). Others require modifications in model architectures or complex post-processings (Ghorbani et al., 2019b; Koh et al., 2020). In contrast, language is inherently understandable by humans and simple to construct. In this work, we interpret the model prediction process by identifying the most influential attributes using language, which provides us meaningful interpretations without pre-processing or post-processing.

**Algorithm Fairness.** Ensuring algorithmic fairness is key to avoiding potential harm to our society (Hovy & Spruit, 2016; Zou & Schiebinger, 2018). Methods for improving the fairness of machine learning algorithms is an ongoing active area of work (Bolukbasi et al., 2016; Sagawa et al., 2020; Sohoni et al., 2020; Ramaswamy et al., 2021; Liu et al., 2021). Among these, a notable solution is to correct data bias, as model bias stems from data bias. In this work, we show that language can be used to correct data bias by generating additional data, hence improving model fairness.

**Limitations.** While our work introduces a novel and effective approach for diagnosing and rectifying visual classifiers, there are additionally important areas for future work. First, since we assume vision classifiers are built using an image-text embedding space trained through multi-modal contrastive learning, our method can also inherit limitations from the contrastive model and pre-training dataset. For example, although we aim to leverage large and general-purpose image-caption datasets in pre-training, the encoders may still not appropriately embed out-of-distribution examples far from what the contrastive model was trained on. Misaligned or inaccurate pre-training data can also affect encoder quality. Additionally, it is challenging to diagnose low-level visual attributes that are difficult to describe in words, such as texture or object orientation (Leclerc et al., 2021). We consider these fruitful directions for future work. Our method will also benefit from improvements in multi-modal contrastive pre-training as these methods are improved.

## 5 CONCLUSION

Our work reveals a valuable advantage of using vision classifiers built on top of multi-modal embedding spaces learned through contrastive learning – the ability to diagnose and rectify the vision classifiers using natural language inputs. We first use a combination of theoretical analysis and experimental findings to verify that cross-modal transferability exists; namely, that text inputs can act as good proxies for image inputs. This then allows us to propose and validate a framework for diagnosing and rectifying vision classifiers using natural language inputs. Our work suggests promising new directions both for achieving reliable and trustworthy computer vision models, and for the use of cross-modal transferability in other problem domains.

## ETHICS STATEMENT

One of the main contributions of our work is an approach for diagnosing and rectifying vision classifiers trained using embeddings from a multi-modal contrastive model. We showcase experimental results on identifying error slices and influential attributes. For example, our method can detect failures caused by the lack of representation of certain races in the training set. In our FairFace experiments, the prediction of gender (i.e., the label "female") given an image was affected by race (e.g., the race "black"). We further show that we can rectify this behavior using our approach. Hence, we see our work as a contribution to the broader community concerned with model accountability and model auditing, and to improving the responsible integration of AI into society.

However, it is also important to be aware of potential negative impacts brought about by our findings. One can imagine an adversary who extends our approach and uses it to their advantage, perhaps reinforcing racial or gender biases by fine-tuning a vision model using biased language prompts. Our work also inherits limitations from the contrastive model and pre-training datasets used to obtain the image and text encoders, as described in the Discussion section of our paper. We hope that this statement raises awareness both of the importance of better model diagnosis and rectification methods and of future directions of work to address limitations and potential negative impacts.

## REPRODUCIBILITY STATEMENT

We provide open-source implementation of our work at `https://github.com/yuhui-zh15/drml`. The implementations will enable researchers to reproduce all the experiments described here as well as run their own analyses on additional multi-modal models and datasets.

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

## OVERVIEW OF APPENDIX

In this appendix, we supplement additional details of theory, datasets, experiments, and baselines.

- In Appendix A, we provide more details about the modality gap geometry, a theoretical proof of cross-modal transferability given the modality gap, additional cross-modal transferability results on MS-COCO and ImageNet.
- In Appendix B, we provide details of four datasets (MS-COCO, Waterbirds, FairFace, and dSpritesV) used in our experiments, including data preprocessing, attributes, and prompts. We also provide the model and experimental details.
- In Appendix C, we provide two baseline methods. First, we present the result using text-to-image generation for model diagnosis, which sometimes fails to generate fidelity images given text prompts. Second, we present the baseline method for slice discovery using DOMINO, which fails when error slices are absent in the dataset.

## A   CROSS-MODAL TRANSFERABILITY

### A.1   MODALITY GAP GEOMETRY

Figure 2 shows the modality gap phenomenon in various multi-modal contrastive learning models, where inputs from two modalities are embedded at arm's length in their shared representation space. This phenomenon is caused by the combined effect of model initialization and optimization. Deep neural networks have the cone effect — encoders will only map inputs to a small cone of the entire representation space. Therefore, two cones will be created for a multi-modal model with two encoders. As a sequence, the modality gap occurs at the initialization stage. During optimization, the contrastive loss will preserve the gap due to mismatched data (Liang et al., 2022).

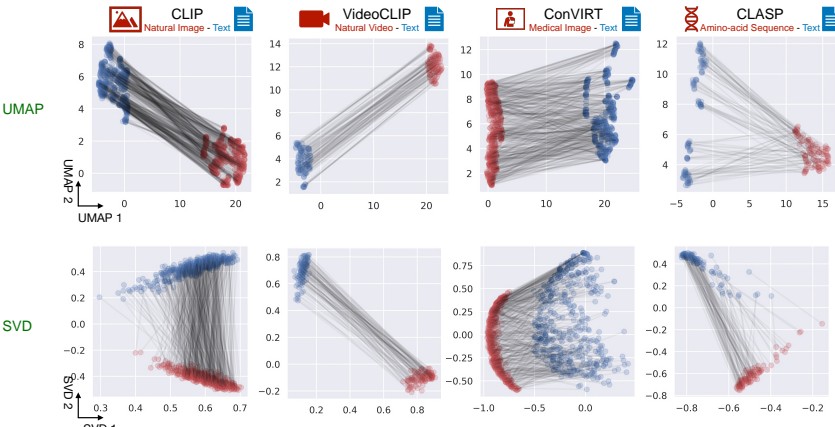

Figure 2: **Modality gap for multi-modal contrastive learning.** Embeddings from two modalities are visualized using UMAP and SVD. Figure credit: Liang et al. (2022).

Figure 3 shows four statistics that reveal important properties of the modality gap geometry.

- *The modality gap approximates a constant vector, particularly at the class level.* We verify this by computing distributions over $\|g\|$ (*magnitude*) and $\cos(g, \mathbb{E}_g[g])$ (*direction*). $g$ is the gap between embeddings of paired data from two modalities.
- *The modality gap is orthogonal to the span of embeddings, and embeddings have zero mean in the subspace orthogonal to the modality gap.* We verify this by computing distributions over $\cos(x - \mathbb{E}_x[x], \mathbb{E}_g[g])$ (*orthogonality*) and $\mathbb{E}_x[x - x^T g'g']_i$ (*center*), where $g' = \mathbb{E}_g[g]/\|\mathbb{E}_g[g]\|$ and $i \in [d]$ denoting $i$-th dimension of the vector.

Based on our theoretical analysis in the next section, these findings suggest that cross-modal transferability is widely established in multi-modal contrastive learning.

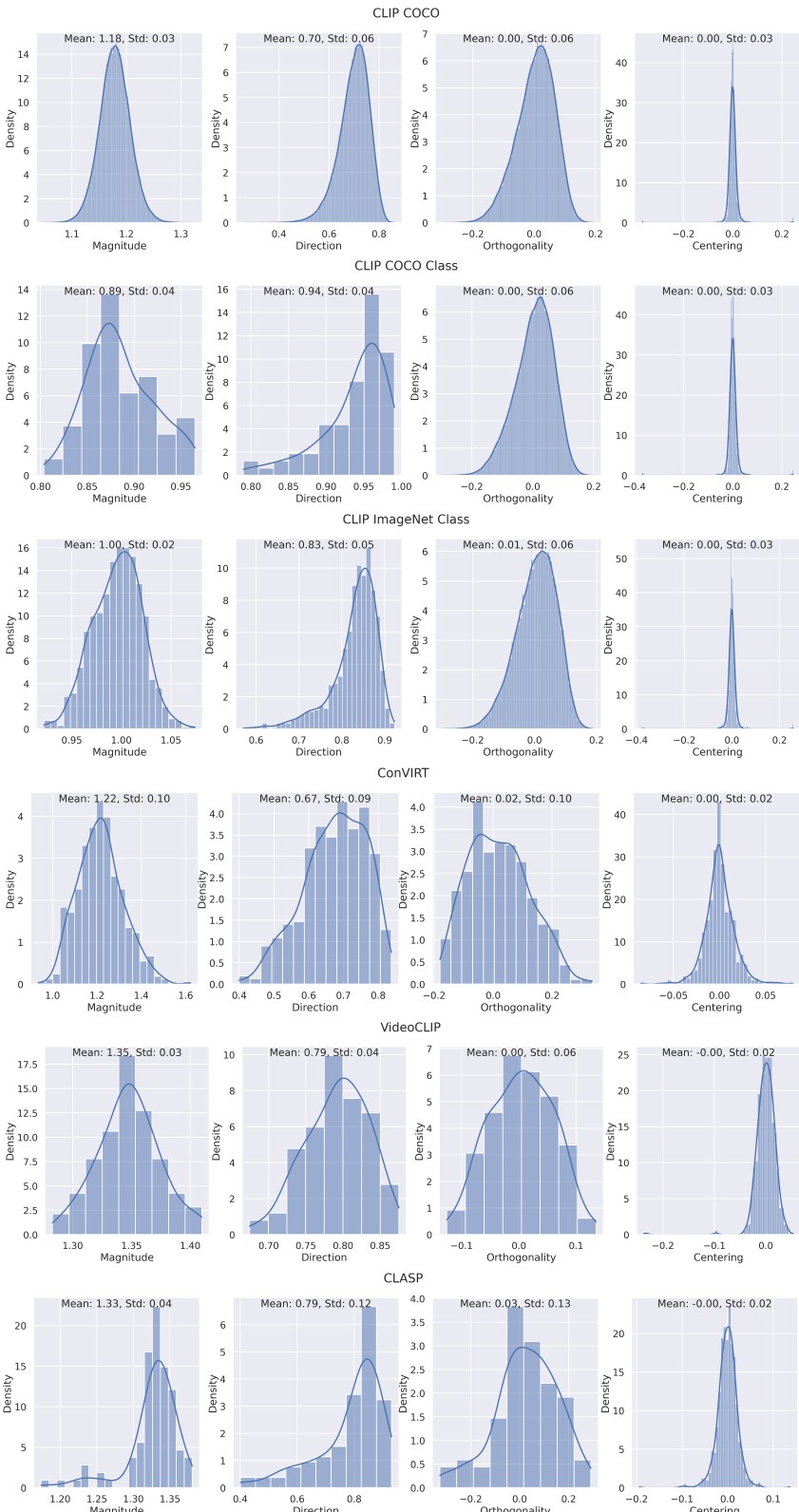

Figure 3: **Geometry analysis of modality gap for various multi-modal contrastive representation spaces.** The modality gap approximates a constant vector, indicated by the magnitude and direction distributions. Modality gap is also orthogonal to the span of embeddings from two modalities, and embeddings' centers for both two modalities are zero vectors in the subspace orthogonal to the gap, indicated by the orthogonality and centering distributions.

## A.2 Theoretical Proof for Cross-Modal Transferability

In this section, we expand and formally discuss what is in section 2.1. We theoretically explain the intriguing cross-modal transferability phenomenon. We explain why the modality gap in the multi-modal representation space does not prevent cross-modal transferability because of the unique geometry of the modality gap.

For class $c \in [|\mathcal{C}|]$, let $\boldsymbol{e}_c \in \{0,1\}^{|\mathcal{C}|}$ be a one-hot vector such that the $c$-th dimension is 1 and other dimensions are 0. We define the following balanced target label vector $\tilde{\boldsymbol{e}}_c := \boldsymbol{e}_c - \mathbb{E}_{c'}[\boldsymbol{e}_{c'}]$, where the expectation is over the distribution of classes on the image domain.

We consider learning a linear function $h_W(\boldsymbol{u}) = \boldsymbol{W}\boldsymbol{u}$, where $\boldsymbol{W} \in \mathbb{R}^{|\mathcal{C}| \times d}$ is the weight matrix and $\boldsymbol{u} \in \mathbb{R}^d$ is the image or text embedding. Given $h_W(\boldsymbol{u})$ and a label $c$, we consider the following quadratic loss:

$$\mathcal{L}_{\text{quad}}(h_W(\boldsymbol{u}), c) = \|h_W(\boldsymbol{u}) - \tilde{\boldsymbol{e}}_c\|_2^2.$$

The following proposition shows that when the gap between image and caption embeddings is the same for all image-caption pairs and is orthogonal to the embedding span for each modality, a linear model trained to minimize the quadratic loss on one modality transfers to the other modality without loss of accuracy.

**Proposition A.1.** *Suppose there exists a gap vector $\boldsymbol{g} \in \mathbb{R}^d$ such that every pair of image embedding $\boldsymbol{x}$ and caption embedding $\boldsymbol{y}$ satisfies $\boldsymbol{g} = \boldsymbol{x} - \boldsymbol{y}$. Suppose the gap $\boldsymbol{g}$ is orthogonal to the span of image features (i.e., $\boldsymbol{g}^T\boldsymbol{x} = \boldsymbol{g}^T\boldsymbol{x}'$ for two image embeddings $\boldsymbol{x}$ and $\boldsymbol{x}'$), and the image features have zero mean in the subspace orthogonal to $\boldsymbol{g}$ (i.e., $\mathbb{E}_x[\Pi_g(\boldsymbol{x})] = \boldsymbol{0}$ where $\Pi_g(\boldsymbol{x})$ projects the vector $\boldsymbol{x}$ to the subspace orthogonal to $\boldsymbol{g}$). Then, for any $\lambda > 0$ and linear function $h_W(\boldsymbol{u})$ that minimizes the regularized quadratic loss $\mathbb{E}_{x,c}[\mathcal{L}_{quad}(h_W(\boldsymbol{x}), c)] + \lambda\|\boldsymbol{W}\|_F^2$, we have that*

$$h_W(\boldsymbol{x}) = h_W(\boldsymbol{y})$$

*Thus, cross-modal transferability happens.*

*Proof of Proposition A.1.* Since $\boldsymbol{g}^T\boldsymbol{x} = \boldsymbol{g}^T\boldsymbol{x}'$ for all image features $\boldsymbol{x}$ and $\boldsymbol{x}'$, we can find a $\tau \in \mathbb{R}$ such that $\boldsymbol{x} = \Pi_g(\boldsymbol{x}) + \tau\boldsymbol{g}$. Notice that

$$\begin{aligned}
\mathbb{E}_{x,c}[\mathcal{L}_{\text{quad}}(h_W(\boldsymbol{x}), c)] &= \mathbb{E}_{x,c}[\|\boldsymbol{W}\boldsymbol{x} - \tilde{\boldsymbol{e}}_c\|_2^2] \\
&= \|\mathbb{E}_x[\boldsymbol{W}\boldsymbol{x}] - \mathbb{E}_c[\tilde{\boldsymbol{e}}_c]\|_2^2 + \mathbb{E}_{x,c}[\|(\boldsymbol{W}\boldsymbol{x} - \tilde{\boldsymbol{e}}_c) - (\mathbb{E}_x[\boldsymbol{W}\boldsymbol{x}] - \mathbb{E}_c[\tilde{\boldsymbol{e}}_c])\|_2^2] \\
&= \|\mathbb{E}_x[\boldsymbol{W}\boldsymbol{x}] - \mathbb{E}_c[\tilde{\boldsymbol{e}}_c]\|_2^2 + \mathbb{E}_{x,c}[\|\boldsymbol{W}\Pi_g(\boldsymbol{x}) - \tilde{\boldsymbol{e}}_c\|_2^2] \\
&= \|\boldsymbol{W}\mathbb{E}_x[\Pi_g(\boldsymbol{x})] + \tau\boldsymbol{W}\boldsymbol{g} - \mathbb{E}_c[\tilde{\boldsymbol{e}}_c]\|_2^2 + \mathbb{E}_{x,c}[\|\boldsymbol{W}\Pi_g(\boldsymbol{x}) - \tilde{\boldsymbol{e}}_c\|_2^2].
\end{aligned}$$

Since $\mathbb{E}_x[\Pi_g(\boldsymbol{x})] = \boldsymbol{0}$ and $\mathbb{E}_c[\tilde{\boldsymbol{e}}_c] = \boldsymbol{0}$, the first term reduces to $\tau^2\|\boldsymbol{W}\boldsymbol{g}\|_2^2$. Notice that the second term in the loss decomposition only involves $\boldsymbol{W}$'s components that are orthogonal to $\boldsymbol{g}$. Thus the minimization of the second term is independent of the minimization of the first term. As a result, any $\boldsymbol{W}$ that minimizes the regularized quadratic loss must satisfy $\boldsymbol{W}\boldsymbol{g} = \boldsymbol{0}$.

For a pair of image and text features $\boldsymbol{x}, \boldsymbol{y}$, since $\boldsymbol{x} - \boldsymbol{y} = \boldsymbol{g}$ and $\boldsymbol{W}\boldsymbol{g} = \boldsymbol{0}$, we have $h_W(\boldsymbol{x}) = h_W(\boldsymbol{y})$, which finishes the proof. $\square$

## A.3 Additional Cross-modal Transferability Results

**MS-COCO.** In the main paper, we only report image-to-text transfer, where we train a classifier on image embeddings and test on text embeddings. Here we report the full results, including text-to-image transfer, in Table 7.

**ImageNet.** In the main paper, we report cross-modal transferability on the MS-COCO dataset. Here we report cross-modal transferability results using the ImageNet dataset (Deng et al., 2009). We split ImageNet validation set into 40K / 10K images for training / evaluation. We apply OpenAI

| Model | Transfer | **In-Modal Evaluation** | | | | | **Cross-Modal Evaluation** | | | | **Consistency$_\uparrow$** |
|---|---|---|---|---|---|---|---|---|---|---|---|
| | | M | Loss$_\downarrow$ | mF1$_\uparrow$ | MF1$_\uparrow$ | M | Loss$_\downarrow$ | mF1$_\uparrow$ | MF1$_\uparrow$ | | |
| Random | - | $\mathcal{I}$ | 0.6939 | 0.0655 | 0.0443 | $\mathcal{T}$ | 0.6938 | 0.0696 | 0.0437 | | 0.8644 |
| | | | | **Default Modality Gap** | | | | | | | |
| Linear | $\mathcal{I} \rightarrow \mathcal{T}$ | $\mathcal{I}$ | 0.0501 | 0.7276 | 0.6790 | $\mathcal{T}$ | 0.1188 | 0.5642 | 0.5429 | | 0.9637 |
| | $\mathcal{T} \rightarrow \mathcal{I}$ | $\mathcal{T}$ | 0.0580 | 0.6983 | 0.6631 | $\mathcal{I}$ | 0.1572 | 0.5320 | **0.4833** | | 0.9527 |
| MLP | $\mathcal{I} \rightarrow \mathcal{T}$ | $\mathcal{I}$ | 0.0480 | 0.7523 | 0.7158 | $\mathcal{T}$ | 0.0888 | 0.6350 | 0.6135 | | 0.9789 |
| | $\mathcal{T} \rightarrow \mathcal{I}$ | $\mathcal{T}$ | 0.0572 | 0.7119 | 0.6826 | $\mathcal{I}$ | 0.0750 | 0.6359 | 0.5929 | | 0.9795 |
| | | | | **Closing Modality Gap** | | | | | | | |
| Linear | $\mathcal{I} \rightarrow \mathcal{T}$ | $\mathcal{I}$ | 0.0498 | 0.7280 | 0.6777 | $\mathcal{T}$ | **0.0719** | **0.6554** | **0.6168** | | **0.9842** |
| | $\mathcal{T} \rightarrow \mathcal{I}$ | $\mathcal{T}$ | 0.0578 | 0.6988 | 0.6628 | $\mathcal{I}$ | **0.0660** | **0.5782** | 0.4767 | | **0.9858** |
| MLP | $\mathcal{I} \rightarrow \mathcal{T}$ | $\mathcal{I}$ | 0.0483 | 0.7495 | 0.7130 | $\mathcal{T}$ | **0.0885** | **0.6503** | **0.6358** | | **0.9806** |
| | $\mathcal{T} \rightarrow \mathcal{I}$ | $\mathcal{T}$ | 0.0573 | 0.7073 | 0.6763 | $\mathcal{I}$ | **0.0685** | **0.6603** | **0.6173** | | **0.9801** |

Table 7: **Cross-modal transferability in multi-modal contrastive representation learning.** We train a classifier using CLIP's image embeddings and test the trained classifier using text embeddings, vice versa, on the MS-COCO multi-label classification dataset. Despite the modality gap, classification boundaries learned from one modality are transferable to another modality. Closing the modality gap further improves cross-modal transferability without harm to in-modal evaluation. Notations: $\mathcal{I}$ - Image, $\mathcal{T}$ - Text, M - Modality, mF1 - Micro F1, MF1 - Macro F1, Random - A randomly initialized linear model.

CLIP's 80 prompts to 1000 ImageNet class names and get 80K texts, and we split them into 64K / 16K for training / evaluation. All the experimental settings are the same as MS-COCO experiments. Results are shown in Table 8.

Again, despite the modality gap, we find that the classification boundaries learned from one modality are transferable to the other modality. When a linear classifier is trained on image embeddings and achieves 70.86% image classification accuracy, directly feeding the text embeddings to the trained classifier achieves 85.24% accuracy. The transfer from text to image is much worse than from image to text, because the texts we used are generated from prompts and thus lack diversity to train a classifier with good decision boundaries. Closing the modality gap improves the transferability in most cases.

| Split | **Image-to-Text** | | **Text-to-Image** | |
|---|---|---|---|---|
| | Linear | MLP | Linear | MLP |
| | **Default Modality Gap** | | | |
| In-Modal Evaluation | 0.7086 | 0.6687 | 0.9974 | 0.9951 |
| Cross-Modal Evaluation | 0.8524 | 0.7758 | 0.4953 | **0.4552** |
| | **Closing Modality Gap** | | | |
| In-Modal Evaluation | 0.7048 | 0.6683 | 0.9978 | 0.9949 |
| Cross-Modal Evaluation | **0.8754** | **0.7892** | **0.5050** | 0.4438 |

Table 8: **Cross-modal transferability in multi-modal contrastive representation learning using the ImageNet dataset.** We split ImageNet validation set 50K images to 40K / 10K for training and evaluation. Texts are generated using OpenAI's 80 prompts multiply by 1000 class names.

## A.4 Theoretical Intuition for Modality Gap and Cross-Modal Transferability

In this section, we provide theoretical insights about the intriguing modality gap and cross-modal transferability phenomenon. We show that after optimizing multi-modal contrastive loss, there is a modality gap between image embeddings and text embeddings, and there is a linear classifier trained on image embeddings that is guaranteed to generalize to text embeddings regardless of the modality gap.

**Basic Notations.** Given an image $x$ and a text $z$, we use an image encoder $f(\cdot)$ and a text encoder $g(\cdot)$ to map them to the shared $D$-dimensional representation space. We denote $f(x) \in \mathbb{R}^D$ as image embedding and $g(z) \in \mathbb{R}^D$ as text embedding. Given $N$ images and $M$ texts, we denote the concatenated image embedding matrix as $F \in \mathbb{R}^{N \times D}$ and concatenated text embedding matrix as $G \in \mathbb{R}^{M \times D}$, which is shown in the following equation:

$$F = \begin{bmatrix} f(x_1)^T \\ ... \\ f(x_N)^T \end{bmatrix} \in \mathbb{R}^{N \times D} \qquad G = \begin{bmatrix} g(z_1)^T \\ ... \\ g(z_M)^T \end{bmatrix} \in \mathbb{R}^{M \times D} \tag{1}$$

**Image-Text Connection Graph.** Given an image $x$ and a text $z$, we denote the probability of $(x, z)$ being an image-text pair as $p(x, z)$. With the probability definition, we have $\sum_{x,z} p(x, z) = 1$. Given $N$ images and $M$ texts, we denote the probability matrix as $P \in \mathbb{R}^{N \times M}$. Note that the probability matrix $P$ is a sparse matrix with most elements as zero, because most image-text pairs are mismatched and cannot be collected (e.g., a cat image with a dog caption). $P$ can be viewed as the adjacency matrix of a bi-particle graph $\mathcal{G} = (\{x, z\}, \{p(x, z)\})$, where all the images and texts are the vertices of the graph and their connection probabilities are the edges. The adjacency matrix of this graph can be written as the following equation:

$$P = \begin{bmatrix} p(x_1, z_1) & ... & p(x_1, z_M) \\ ... & ... & ... \\ p(x_N, z_1) & ... & p(x_N, z_M) \end{bmatrix} \tag{2}$$

We have the following theorem which shows the connection between multi-modal contrastive learning with the partitioning of the connection graph defined above. This result can be viewed as a simple generalization of the results in HaoChen et al. (2021) to the multi-modal setting.

**Theorem 1** (Equivalence of Multi-modal Contrastive Loss and Graph Partitioning). *With the mild assumption that every image and every text pair has equal presence probability* $\forall x : p_x = \sum_z p(x, z) = \frac{1}{N}$, $\forall z : p_z = \sum_x p(x, z) = \frac{1}{M}$, *minimizing the multi-modal contrastive loss in Equation 3 is equivalent to minimizing* $\|P - FG^T\|_F^2$:

$$\mathcal{L} = -2\mathbb{E}_{x,z}\left[f(x)^T g(z)\right] + NM\mathbb{E}_{x \sim P_x, z \sim P_z}\left[\left(f(x)^T g(z)\right)^2\right] \tag{3}$$

**Proof of Theorem 1.** The following equation proves Theorem 1:

$$\begin{aligned}
&\min \|P - FG^T\|_F^2 \\
&= \min \sum_{x,z} -2p(x,z)f(x)^T g(z) + \sum_{x,z}\left(f(x)^T g(z)\right)^2 + \sum_{x,z} p(x,z)^2 \\
&= \min \sum_{x,z} -2p(x,z)f(x)^T g(z) + \sum_{x,z}\left(f(x)^T g(z)\right)^2 \\
&= \min \sum_{x,z} -2p(x,z)f(x)^T g(z) + NM \sum_{x,z} \frac{1}{N}\frac{1}{M}\left(f(x)^T g(z)\right)^2 \\
&= \min -2\mathbb{E}_{x,z}\left[f(x)^T g(z)\right] + NM\mathbb{E}_{x \sim P_x, z \sim P_z}\left[\left(f(x)^T g(z)\right)^2\right]
\end{aligned} \tag{4}$$

**Connection to CLIP Contrastive Loss.** Given $N$ images and $N$ texts, with the assumption $\forall i, j : p(x_i, z_j) = \frac{1}{N}\mathbb{1}[i = j]$, the CLIP contrastive loss is shown in Equation 5 and is very similar to the contrastive loss in Equation 3.

$$\mathcal{L}_{\text{CLIP}}$$

$$= \frac{1}{N} \sum_i \left[ -\log \frac{\exp\left(f(x_i)^T g(z_i)\right)}{\sum_j \exp\left(f(x_i)^T g(z_j)\right)} - \log \frac{\exp\left(f(x_i)^T g(z_i)\right)}{\sum_j \exp\left(f(x_j)^T g(z_i)\right)} \right]$$

$$= \frac{1}{N} \sum_i \left[ -2f(x_i)^T g(z_i) + \log \sum_j \exp\left(f(x_i)^T g(z_j)\right) + \log \sum_j \exp\left(f(x_j)^T g(z_i)\right) \right]$$

$$= \frac{-2}{N} \sum_i f(x_i)^T g(z_i) + \frac{1}{N} \left( \sum_i \log \sum_j \exp\left(f(x_i)^T g(z_j)\right) + \sum_i \log \sum_j \exp\left(f(x_j)^T g(z_i)\right) \right)$$

$$= -2 \mathbb{E}_{x,z}\left[ f(x)^T g(z) \right] + \mathbb{E}_{x \sim P_x, z \sim P_z}\left[ \log \sum_{z'} \exp(f(x)^T g(z')) + \log \sum_{x'} \exp(f(x')^T g(z)) \right]$$

$$(5)$$

Let us now consider the modality gap phenomenon in the above mentioned contrastive learned representation space.

**Proposition 1** (Modality Gap). *After optimizing the multi-modal contrastive loss defined in Equation 3, image embedding and text embeddings will be separated in the shared presentation space, causing the modality gap phenomenon.*

**Proof of Proposition 1.** Since optimizing the multi-modal contrastive loss defined in Equation 3 is equivalent to minimizing $\|P - FG^T\|_F^2$, achieving the minima of $\|P - FG^T\|_F^2$ does not indicate image embeddings $F$ and text embeddings $G$ are close in the embedding space. For instance, since for any scalar $c > 0$, we can scale $F$ by a factor of $c$ and $G$ by a factor of $1/c$ without changing the contrastive loss, we know that there exist many solutions that both achieve the minimal loss and also exhibit modality gap.

Now we consider the cross-modal transferability phenomenon on downstream classification problems using the above mentioned contrastive learned representations. We first introduce the following notations for the downstream task's labels.

**Labels of Images and Texts.** Given an image $x$ with the label $y_x \in [C]$, where $C$ is the number of classes, we denote its one-hot representation as $e_{y_x} = \left[ \mathbb{1}[y_x = 1], ..., \mathbb{1}[y_x = C] \right]^T$. Given $N$ images, we denote the image label matrix as $Y_x = \left[ e_{y_{x_1}}^T, ..., e_{y_{x_N}}^T \right] \in \{0, 1\}^{N \times C}$. We use similar notations $y_z, e_{y_z}, Y_z$ for the text.

We make the following assumption which says the label of a text can be predicted by the labels of the images that it is paired with.

**Assumption 1.** *With the definition of connection probablity matrix $P$, image label matrix $Y_x$ and text label matrix $Y_z$, we have $P^T Y_x = \frac{1}{M} Y_z$.*

**Intuition of Assumption 1.** In most realistic settings, the connection between an image and a text $p(x, z) > 0$ if $y_x = y_z$ and $p(x, z) = 0$ if $y_x \neq y_z$. Therefore, $\forall z : \sum_x p(x, z) e_{y_x} = \frac{1}{M} e_{y_z}$. The following equation proves the matrix form:

$$P^T Y_x = \begin{bmatrix} p(x_1, z_1) & ... & p(x_N, z_1) \\ ... & ... & ... \\ p(x_1, z_M) & ... & p(x_N, z_M) \end{bmatrix} \begin{bmatrix} e_{y_{x_1}}^T \\ ... \\ e_{y_{x_N}}^T \end{bmatrix} = \begin{bmatrix} \sum_i p(x_i, z_1) e_{y_{x_i}}^T \\ ... \\ \sum_i p(x_i, z_M) e_{y_{x_i}}^T \end{bmatrix} = \frac{1}{M} \cdot \begin{bmatrix} e_{y_{z_1}}^T \\ ... \\ e_{y_{z_M}}^T \end{bmatrix} = \frac{1}{M} \cdot Y_z$$

$$(6)$$

**Proposition 2** (Cross-modal Transferability). *After optimizing the multi-modal contrastive loss introduced in Equation 3 with infinite encoder dimension $D = \infty$, we can find a linear classifier that only uses image representations but is transferable to text representations. More specifically, the weight of the linear layer is $MF^T Y_x \in \mathbb{R}^{D \times C}$, which can be viewed as a class-mean classifier.*

**Proof of Proposition 2.** After optimizing the contrastive loss, we have $P = FG^T$ (Theorem 1). With $P^T Y_x = \frac{1}{M} Y_z$ (Assumption 1), we have $G(MF^T Y_x) = M(FG^T)^T Y_x = MP^T Y_x = Y_z$. The following equation explains why $MF^T Y_x$ is a class-mean classifier:

$$
MF^T Y_x = M \begin{bmatrix} f(x_1) & ... & f(x_N) \end{bmatrix} \begin{bmatrix} e_{y_{x_1}}^T \\ ... \\ e_{y_{x_N}}^T \end{bmatrix} = M \begin{bmatrix} \sum_x f(x)\mathbb{1}[y_x = 1] & ... & \sum_x f(x)\mathbb{1}[y_x = C] \end{bmatrix} \tag{7}
$$

Intuitively, the class-mean linear head is very similar to the trained linear head. For instance, learning the linear head with one step of gradient descent starting from zero initialization would recover the class-mean linear head. We study the class-mean linear head here because it's more amenable to theoretical analysis. This provides intuition why cross-modal transferability can be achieved regardless of the modality gap.

## B  DATASETS AND EXPERIMENTAL DETAILS

In this section, we report details of four datasets: MS-COCO (Lin et al., 2014), Waterbirds (Sagawa et al., 2020), FairFace (Karkkainen & Joo, 2021), and dSpritesV (Matthey et al., 2017), and details of two major experiments.

### B.1  DATA PRE-PROCESSING

**MS-COCO.** We follow the standard MS-COCO dataset split, which includes 118K / 5K images for training / validation. Each image is annotated with multiple objects from 80 categories and five human-written captions. We randomly select one caption from five captions. Therefore, we have 118K / 5K image-caption pairs with multiple labels for training / validation.

**Waterbirds.** We follow the standard Waterbirds dataset split, which includes 4.8K / 1.2K images for training / validation. Data samples can be viewed in Figure 6 and 7.

**FairFace.** We resample the training set using the demographics from the state of Montana, which includes 92.8% White, 6.4% Indian, 0.5% Asian, and 0.3% Black. The final dataset contains 17K / 11K images for training / validation. Data samples can be viewed in Figure 6 and 7.

**dSpritesV.** We use our own scripts to reproduce a variant of the dSprites dataset and name it dSpritesV. We use six colors (red, pink, orange, green, cyan, blue), four locations (upper left, upper right, lower left, lower right), and three sizes (small, medium, large) to create triangles and squares with a scale ranging from 0.8 to 1.2. Each attribute is uniformly sampled, and we synthesize 10K images. We only use 80% orange triangle and 80% green square for training. Finally, it has 1.3K / 8.7K images for training / validation. Data samples can be viewed in Figure 6 and 7.

### B.2  ATTRIBUTES

Here we list the known attributes during the data collection of the three datasets. We do not cherry-pick attributes and just use these attributes for our experiments.

**Waterbirds.** Two attributes are used: species (200 values) and places (4 places).

**FairFace.** Three attributes are used: races (7 values), ages (9 values), and genders (2 values).

**dSpritesV.** Three attributes are used: colors (6 values), size (3 values), and shapes (2 values).

### B.3  PROMPT ENGINEERING

We use the prompt engineering techniques proposed in CLIP (Radford et al., 2021) for our experiments.

**Waterbirds.** We use "{species}, {place}." as the raw prompt, and "a photo of a {species} in the {place}." as the engineered prompt. Therefore, we can generate $200 \times 4 = 800$ text inputs.

**FairFace.** We use "{age adjective}, {race}, {gender}." as the raw prompt, and "a photo of a {race} {age adjective} {gender}." as the engineered prompt. Therefore, we can generate $7 \times 9 \times 2 = 126$ text inputs. The age adjectives are infant (0-2), little (3-9), teenage (10-19), young (20-29), adult (30-39), middle-aged (40-49), senior (50-59), elderly (60-69), and very old (more than 70).

**dSpritesV.** We use "{size}, {color}, {shape}." as the raw prompt, and "{size} {color} {shape}." as the engineered prompt. Therefore, we can generate $3 \times 6 \times 2 = 36$ text inputs.

## B.4 PROMPT ENSEMBLE

We use OpenAI CLIP's 80 prompts (Radford et al., 2021) to augment text inputs by 80 times. Parts of them are shown in Figure 4.

## B.5 EXPERIMENTAL DETAILS

**Model Details.** Unless explicitly stated, we use CLIP (ViT-B/32) for all experiments, encoding images and texts in the same 512-dimensional space. Linear layer maps input dimension 512 to the number of classes. Multi-layer perception uses the hidden size as 512.

**Cross-modal Transferability Training Details.** For each image-caption pair, we use CLIP's image and text encoder (Radford et al., 2021) to get its image embedding and text embedding. We do not use image augmentation techniques during training and inference. We train the linear model or multi-layer perception for 25 epochs using the Adam optimizer with a fixed learning rate of 0.001. During training, CLIP's image and text encoder are fixed. We pick the best model based on the lowest validation loss on the training modality.

**Classifiers Training Details.** For each image in the dataset, we use CLIP's image encoder (Radford et al., 2021) to get its image embedding. We do not use image augmentation techniques during training and inference. We train the linear model or multi-layer perception for 25 epochs using the Adam optimizer with a fixed learning rate of 0.001. During training, CLIP's image encoder is fixed. We pick the best model based on the lowest validation loss on images.

**Model Rectification Training Details.** For each text from error slices generated by attribute composition and prompt engineering, we use CLIP's text encoder (Radford et al., 2021) to get its text embedding. We continue training the pre-trained linear model or multi-layer perception for 10 epochs using the Adam optimizer with a fixed learning rate of 0.001. During training, CLIP's text encoder is fixed. We pick the best model based on the lowest validation loss on texts.

## B.6 DISCUSSION: IS USING ATTRIBUTES AN APPROPRIATE CHOICE FOR MODEL DIAGNOSIS?

In this work, we heavily rely on attributes for model diagnosis and rectification. A natural question is what is the rationale for using attributes in these processes, and how much will attribute selection affect the validity of our method.

We first clarify that in our experiments, we did not cherry-pick attributes and just used the known attributes from the data curation process of the three datasets. More broadly, we believe attributes are useful for model diagnosis, because it is not hard to define a meaningful set of attributes given a specific task; there are many known attributes given any dataset, and sometimes there may be specific attributes of interest for which we wish to test model vulnerability. For example, for a self-driving car application, we can easily come up with attributes such as weather, traffics, pedestrians, buildings, etc. For any class in ImageNet classification, such as guitar, it is straightforward to think about its color, material, location, size, etc. We agree that an initial set of chosen attributes may not be perfect for reflecting all the essential errors, and better attribute selection can reveal more model

vulnerabilities. Therefore, this process can be improved with human involvement to iteratively design better attributes based on the model feedback, which can be useful for future works.

While attribute selection is important and may require human involvement, our method is still very useful because we provide an easy way to test the model under many cases. Like software testing, there is generally no free lunch for model diagnosis, and it is impossible to design a general diagnosis framework for any task without any prior knowledge. We have already significantly reduced the diagnosis cost compared to previous works. Previous works all assume a large collection of labeled images is available for model testing, which is unrealistic given the extreme difficulty of collecting diverse image inputs and the cost of data annotation. Our method instead provides a way to test sensitive attributes for which you may even have no image data. For any specific task, it is always much easier to come up with a meaningful set of attributes and then generate a large collection of novel text inputs by combining different attributes than collecting corresponding images, thanks to the easy-to-manipulate and compositional nature of the language modality. More importantly, the combination of defined attributes naturally defines human-interpretable data slices, whereas image-based slice discovery methods do not directly provide a text summary of the error slice.

Finally, we hope to clarify that one of the main contributions of our work is to theoretically and empirically demonstrate a pervasive phenomenon in multi-modal contrastive learning —— cross-modal transferability, which allows texts to be effective proxies for images. Our method performance, in terms of correlation strength between model performances on images and corresponding texts, is independent of attribute selection. It is just more errors can be discovered with more human involvement in this process. Moreover, it is possible to collect a large set of text inputs in a different way to diagnose vision models instead of using the attribute-based combination. For example, one may be able to prompt large language models such as GPT-3 in a few-shot fashion to generate a large set of descriptions of certain classes, and then feed these inputs into vision models for diagnosis. We leave this to future work. Overall, diagnosing vision models using text modality is much more desirable than image modality, because language enables us to easily generate realistic and diverse inputs with better control and manipulation.

## C  BASELINES

### C.1  LANGUAGE-BASED VISION MODEL DIAGNOSIS BASELINE: TEXT-TO-IMAGE GENERATION

In Figure 5, we show the baseline method to diagnose vision models using language — text-to-image generation. The method generates real images to test models using the text-to-image generation model.

From the results, we can understand why this baseline is worse than our method, which does not require explicitly generating images. While significant progress has been made in the text-to-image generation field, state-of-the-art text-to-image generation models (Rombach et al., 2022) still fail to generate fidelity images.

In addition, text-to-image generation is computationally expensive, requiring thousands of more computations than our approach.

### C.2  ERROR SLICE DISCOVERY BASELINE: DOMINO

In Figure 6 and 7, we show the discovered error slices using the baseline slice discovery method, DOMINO (Eyuboglu et al., 2022).

In real-world applications, it is unrealistic to assume a large set of labeled images from different distributions is available. Therefore, the most critical challenge for slice discovery is *data*. In this work, we circumvent the data challenge by using language to synthesize extensive test examples.

### C.3  MODEL RECTIFICATION BASELINES: GDRO AND JTT

Our approach rectifies model misbehaviors by correcting the data bias. Another series of methods to tackle these errors are robust training techniques, such as GDRO (Sagawa et al., 2020) and JTT (Liu

et al., 2021), which explicitly optimizes each slice's performance during training. Compared to them, one of the distinct advantages of our approach is that we do not require any visual data during the rectification process. Both GDRO and JTT require image data present in the training set. Therefore, they cannot fix errors on unseen data. GDRO even requires attribute annotations on images, which is highly time-consuming and cost-prohibitive for most real-world applications. Moreover, our method can also be combined with robust training techniques when image data and attribute annotations are available, and we leave this to future work.

Here we provide implementation details of GDRO and JTT:

**GDRO.** We reproduce GDRO on our datasets by adopting the official GDRO loss implementation to our code base. We use all the same hyperparameters they use in the paper, where important hyperparameters include $l_2$ penalty strength $\alpha = 0.2$ and group adjustment $\gamma = 0.1$. We train a linear classifier for 25 epochs using the Adam optimizer with a fixed learning rate of 0.001. During training, CLIP's image encoder is fixed. We pick the best model based on the lowest validation loss.

**JTT.** We reproduce JTT on our datasets by implementing the algorithm by ourselves. We use all the same hyperparameters they use in the paper. We also perform a hyperparameter search on the upsampling weight $\lambda_{up} \in \{5, 20, 50\}$, which is a very important hyperparameter based on the paper. The best $\lambda_{up}$ is 20 for Waterbirds and 5 for FairFace. We train a linear classifier for 25 epochs using the Adam optimizer with a fixed learning rate of 0.001 for round 1 and round 2. During training, CLIP's image encoder is fixed. We pick the best model based on the lowest validation loss.

```python
openai_imagenet_template = [
    lambda c: f"a bad photo of a {c}.",
    lambda c: f"a photo of many {c}.",
    lambda c: f"a sculpture of a {c}.",
    lambda c: f"a photo of the hard to see {c}.",
    lambda c: f"a low resolution photo of the {c}.",
    lambda c: f"a rendering of a {c}.",
    lambda c: f"graffiti of a {c}.",
    lambda c: f"a bad photo of the {c}.",
    lambda c: f"a cropped photo of the {c}.",
    lambda c: f"a tattoo of a {c}.",
    lambda c: f"the embroidered {c}.",
    lambda c: f"a photo of a hard to see {c}.",
    lambda c: f"a bright photo of a {c}.",
    lambda c: f"a photo of a clean {c}.",
    lambda c: f"a photo of a dirty {c}.",
    lambda c: f"a dark photo of the {c}.",
    lambda c: f"a drawing of a {c}.",
    lambda c: f"a photo of my {c}.",
    lambda c: f"the plastic {c}.",
    lambda c: f"a photo of the cool {c}.",
    lambda c: f"a close-up photo of a {c}.",
    lambda c: f"a black and white photo of the {c}.",
    lambda c: f"a painting of the {c}.",
    lambda c: f"a painting of a {c}.",
    lambda c: f"a pixelated photo of the {c}.",
    lambda c: f"a sculpture of the {c}.",
    lambda c: f"a bright photo of the {c}.",
    lambda c: f"a cropped photo of a {c}.",
    lambda c: f"a plastic {c}.",
    lambda c: f"a photo of the dirty {c}.",
    lambda c: f"a jpeg corrupted photo of a {c}.",
    lambda c: f"a blurry photo of the {c}.",
    lambda c: f"a photo of the {c}.",
    lambda c: f"a good photo of the {c}.",
    lambda c: f"a rendering of the {c}.",
    lambda c: f"a {c} in a video game.",
    lambda c: f"a photo of one {c}.",
    lambda c: f"a doodle of a {c}.",
    lambda c: f"a close-up photo of the {c}.",
    lambda c: f"a photo of a {c}.",
    lambda c: f"the origami {c}.",
    lambda c: f"the {c} in a video game.",
    lambda c: f"a sketch of a {c}.",
    lambda c: f"a doodle of the {c}.",
    lambda c: f"a origami {c}.",
    lambda c: f"a low resolution photo of a {c}.",
    lambda c: f"the toy {c}.",
    lambda c: f"a rendition of the {c}.",
    lambda c: f"a photo of the clean {c}.",
    lambda c: f"a photo of a large {c}.",
    lambda c: f"a rendition of a {c}.",
    lambda c: f"a photo of a nice {c}.",
    lambda c: f"a photo of a weird {c}.",
    lambda c: f"a blurry photo of a {c}.",
    lambda c: f"a cartoon {c}.",
    lambda c: f"art of a {c}.",
    lambda c: f"a sketch of the {c}.",
    lambda c: f"a embroidered {c}.",
    lambda c: f"a pixelated photo of a {c}.",
]
```

Figure 4: A subset of the 80 prompts form OpenAI we use to augment text inputs for prompt ensembling.

| Text prompt | Well generated examples | Poorly generated examples |
|---|---|---|

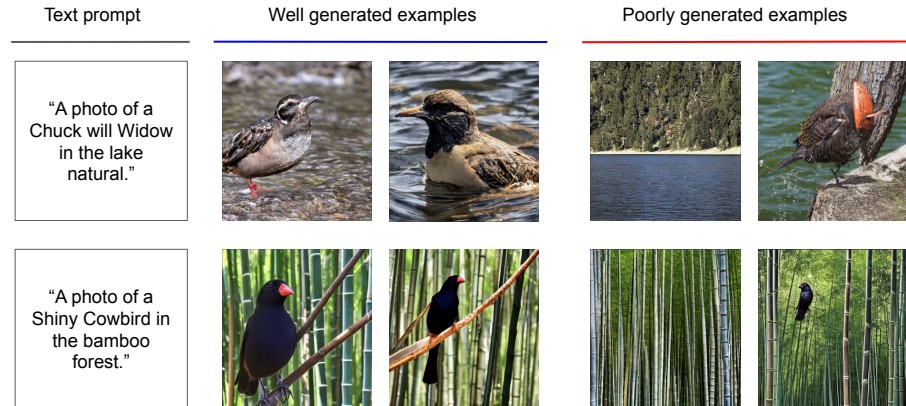

Well-generated examples (columns 2-3) are realistic and correctly reflect the bird species and background location described in the text prompts. Poorly generated images either do not include the bird species (column 4) or are noticeably unnatural (column 5). Specifically, the top image in column 5 includes a bird with lobster claws as its head — a rather unusual phenomenon in the real world

| Text prompt | Well generated examples | Poorly generated examples |
|---|---|---|

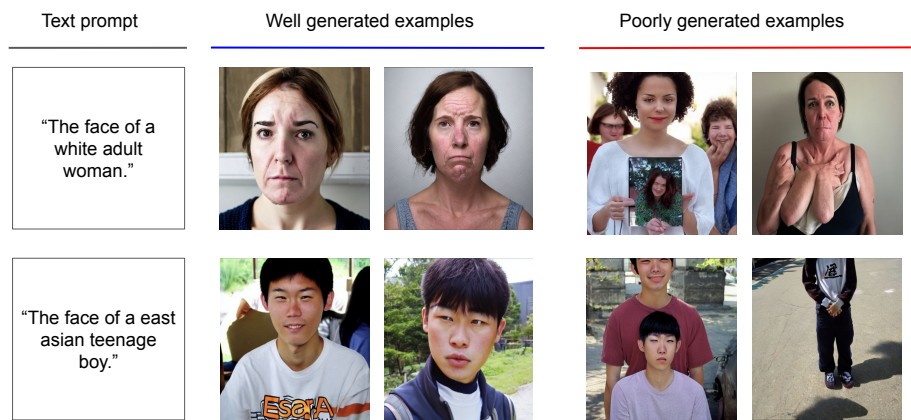

Well-generated examples (columns 2-3) are realistic and correctly reflect the ethnicity and age group described in the text prompts. Poorly generated images include more than one person (column 4), do not include the person's face (column 5 bottom), or are noticeably unrealistic (column 5 top). For instance, the top image in column 5 includes a woman with arms protruding out of her chest, which is rare in the real world.

| Text prompt | Well generated examples | Poorly generated examples |
|---|---|---|

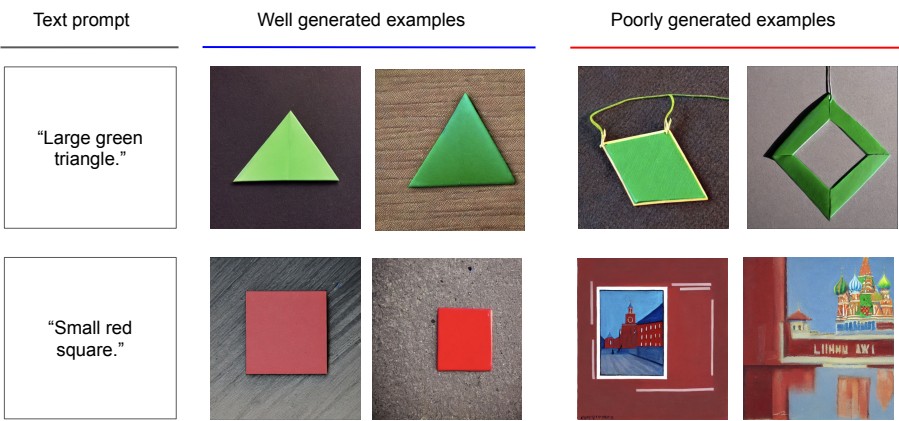

Well-generated examples (columns 2-3) correctly reflect the shape and color described in the text prompt. Poorly generated images either include incorrect shapes (column 4-5 top) or are misinterpreted due to polysemy (column 4-5 bottom). For instance, the phrase "red square" was misinterpreted by our model as a historic site in Moscow.

Figure 5: Text-to-image generation results on Waterbirds, FairFace, and dSpitesV using the state-of-the-art generation model (Rombach et al., 2022).

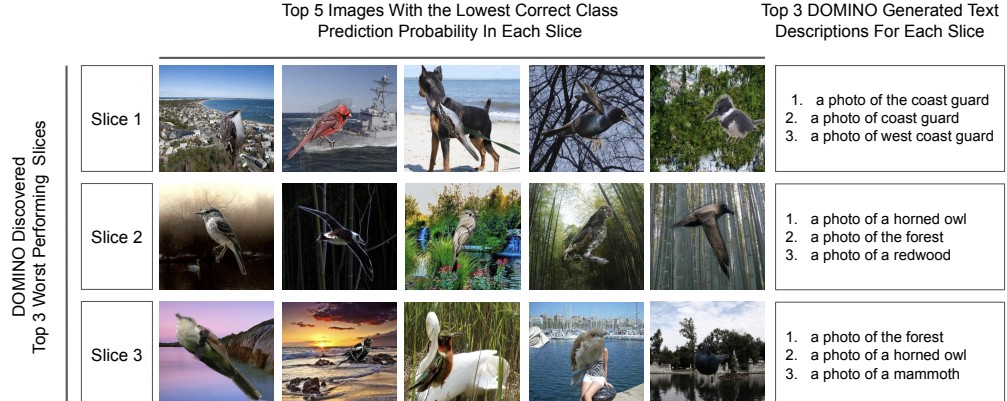

For in-distribution Waterbird, the text descriptions generated from DOMINO do not elucidate which attributes are causing misclassifications. In addition, some of the descriptions are incorrectly generated. For instance, "coast guards" are not present in any of the photos in Slice 1.

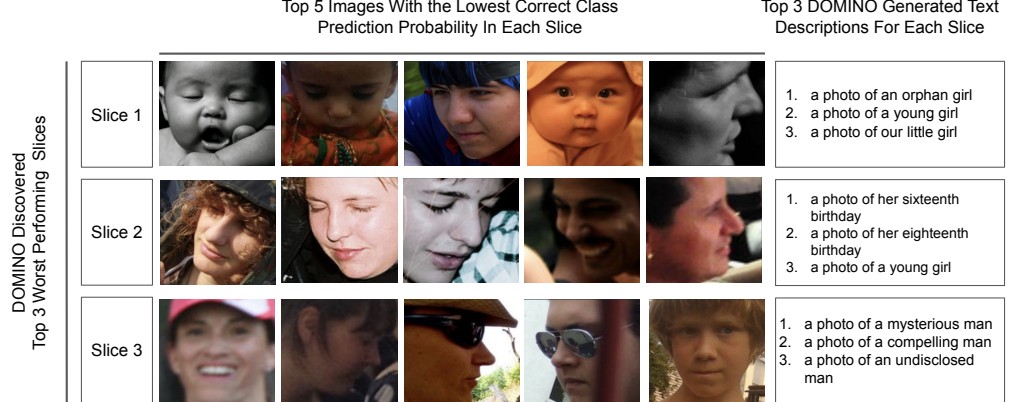

For in-distribution Fairface, DOMINO is unable to discover the correct error slices. Instead of slices for minority groups, DOMINO discovered slices for the prevalent ethnic group in the dataset. Furthermore, the generated descriptions failed to include the key attribute — race.

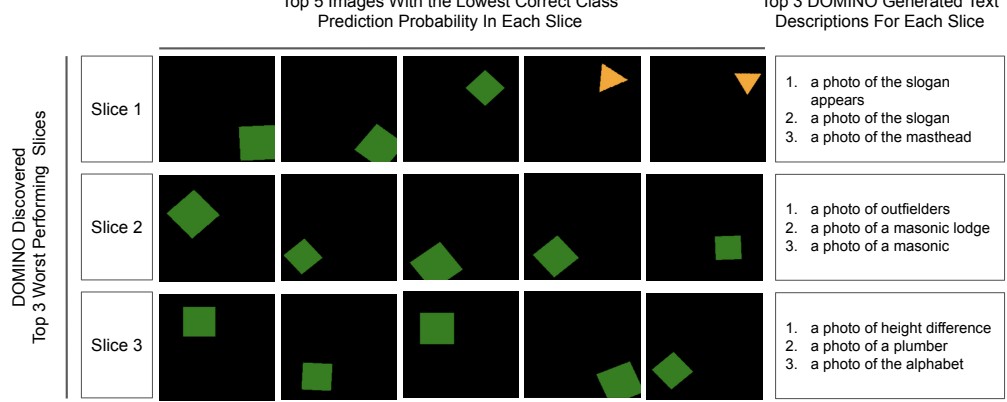

For in-distribution dSpritesV, DOMINO failed to discover the most critical error slices - slices with spurious correlations (orange squares & green triangles) and slices with unseen data (pink triangle). The generated descriptions also do not reflect images in the slices.

Figure 6: Discovered error slices on **in-distribution** Waterbirds, FairFace, and dSpitesV datasets using the state-of-the-art slice discovery method DOMINO (Eyuboglu et al., 2022). DOMINO is unable to discover most of the top error slices. The generated text descriptions often do not reflect images in the slice accurately.

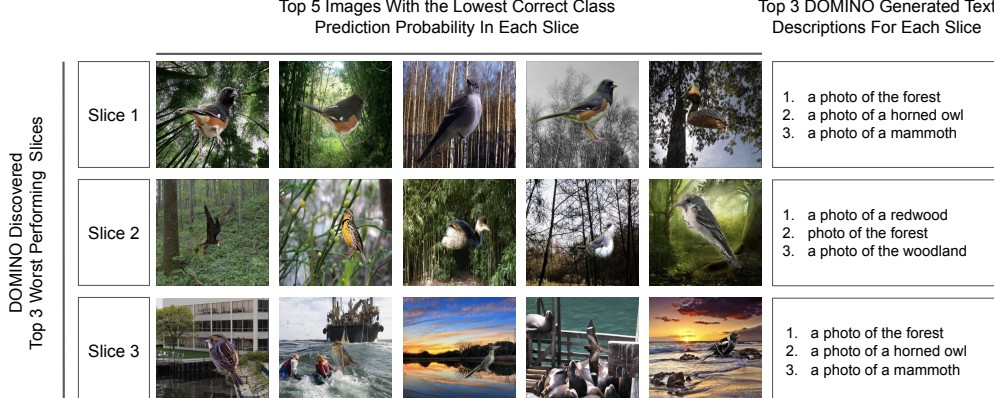

For out-of-distribution Waterbird, DOMINO was unable to generate text that clearly defines the source of misclassification for each slice. Moreover, the some of the descriptions do not accurately describe the images. For example, "mammoths" are never included in any images.

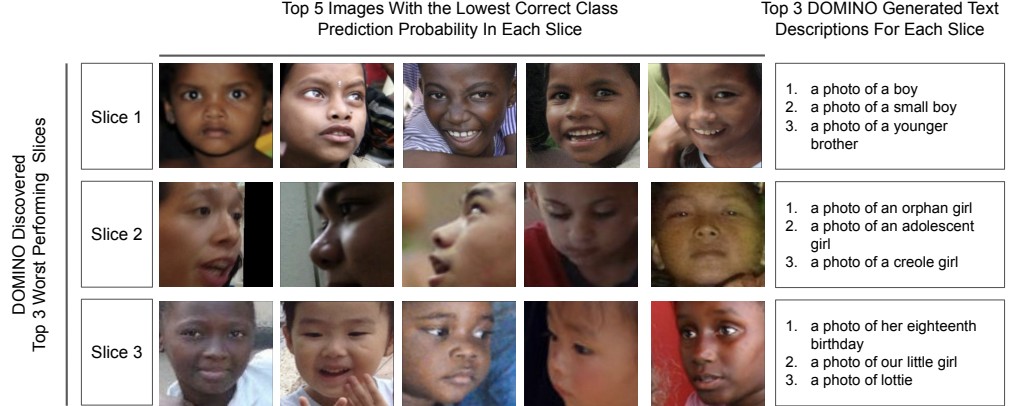

For out-of-distribution Fairface, DOMINO is able to discover the some correct error slices. However, the generated descriptions failed to provide any insights as to why the model fail for these slices. Specifically, the generated descriptions do not include keywords for the race attribute.

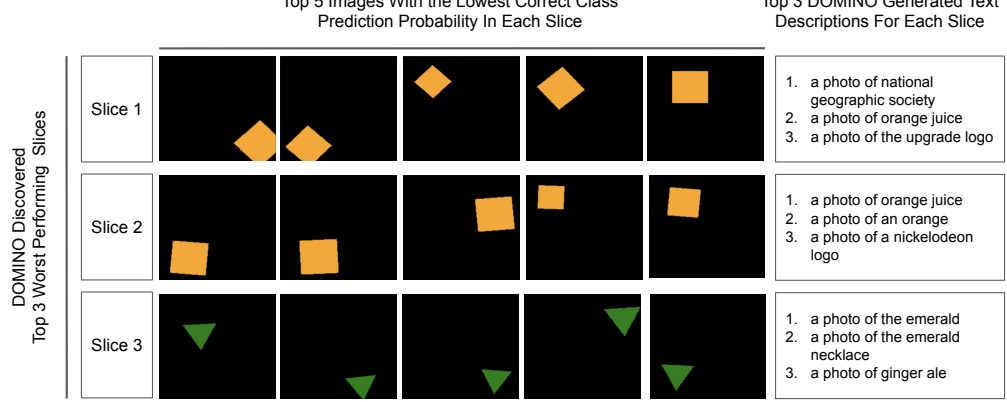

For out-of-distribution dSpritesV, DOMINO is able to discover slices with spurious correlation (orange squares & green triangles). However, the generated text descriptions do not include attributes that contribute to the spurious correlation. Additionally, DOMINO did not discover slices with unseen data (pink triangles).

Figure 7: Discovered error slices on **out-of-distribution** Waterbirds, FairFace, and dSpitesV datasets using the state-of-the-art slice discovery method DOMINO (Eyuboglu et al., 2022). DOMINO was able to capture some, but not all, error slices. Furthermore, artificially generating out-of-distribution data for evaluation remains challenging in real-world settings.

