# OpenReview forum: "Diagnosing and Rectifying Vision Models using Language"
_ICLR.cc/2023/Conference — ICLR 2023 poster_

### Official Review · Reviewer_n4R4 · 2022-10-22

**Confidence:** 3
**Correctness:** 3
**Technical Novelty And Significance:** 3
**Empirical Novelty And Significance:** 3
**Recommendation:** 6

**Clarity, Quality, Novelty And Reproducibility:**

Hi, I'm currently on a medical leave and won't be able to perform ICRL review duties. sorry for the late notice.

**Strength And Weaknesses:**

Hi, I'm currently on a medical leave and won't be able to perform ICRL review duties. sorry for the late notice.

**Summary Of The Paper:**

Hi, I'm currently on a medical leave and won't be able to perform ICRL review duties. sorry for the late notice.

**Summary Of The Review:**

Hi, I'm currently on a medical leave and won't be able to perform ICRL review duties. sorry for the late notice.

---

### Official Review · Reviewer_Wtjn · 2022-10-24

**Confidence:** 4
**Correctness:** 3
**Technical Novelty And Significance:** 4
**Empirical Novelty And Significance:** 3
**Recommendation:** 6

**Clarity, Quality, Novelty And Reproducibility:**

- Clarity and quality

The results show how the embedding structure can be exploited on three cross-modal tasks. However, I found their analysis a little short: for instance, the literature about feature importance is quite large, even in a multimodal setting [1] and the approach should have been compared to others.

[1] Joshi, G., Walambe, R., & Kotecha, K. (2021). A review on explainability in multimodal deep neural nets. IEEE Access, 9.

The geometric structure is only verified empirically. As such, more rigorous statistical testing could have been developed, for instance a chi-2 to test if the modality gap is constant.
About the “constant” modality gap: is there a theoretical justification of it? How general is this phenomenon? Is it a consequence of using contrastive learning ?

- Novelty

Analyzing the geometrical structure of large scale multimodal embeddings, as far as I know, has been seldom done, with the exception of (Liang et al., 2022) from which the paper is built.

- Reproducibility

An anonymous GitHub link with the code is provided.


**Details Of Ethics Concerns:**

The paper discusses the potential use of multimodal embedding for model accountability or auditing, but also the risk of using it to generate biases.


**Strength And Weaknesses:**

== Strengths ==
- The paper is well written and easy to follow.
- Well justified exploitation of multimodal embedding and of their empirical properties
- Application on three rather original tasks using cross-modal transferability

== Weaknesses ==
- The findings (structure of the embeddings) are empirical and not justified by the (contrastive) learning scheme.
- The three tasks are more illustrations of the multimodal embedding properties than complete studies: they are not fully developed or analyzed.


**Summary Of The Paper:**

The proposed approach exploits an empirical property of the latent space of multimodal embedding (essentially image & text) revealed in (Liang et al., 2022): the fact that the two modalities are embedded in different parts in the latent space, inducing a modality gap. The paper extends the results by empirically stating that the two embedded modalities are correlated by a quasi constant shift, easing cross-modal transferability, and that the data manifolds are orthogonal to the modality gap.The multi-modal correlation is used to analyze the quality of vision based prediction on three tasks: discovering hard examples described by sets of attributes (error slices), scoring attributes by their impact on performance using Shapley  values, and by fine tuning a given model using text data augmentation.


**Summary Of The Review:**

The paper pursues the analysis of the modality gap initiated by (Liang et al., 2022) and discusses another finding: the almost constant distribution of this modality gap and its geometry with respect to the data manifolds. It proposes three original tasks that exploit the results. They are, however, mostly empirical, and could have been verified with better statistical justification; the analysis of the tasks is also not very thorough.

---

> ### Author Response · Authors · 2022-11-10
> **[1/2] Response to Reviewer Wtjn**
>
> We thank Reviewer Wtjn for their positive comments and for providing thoughtful feedback on our work. We address Reviewer Wtjn’s concerns below.
>
> **More explanations about modality gap geometry**
>
> > The geometric structure is only verified empirically. As such, more rigorous statistical testing could have been developed, for instance a chi-2 to test if the modality gap is constant.
> >
>
> Thank you for your question. The chi-2 test is not appropriate for our case because it is designed for categorical data, while our data are continuous. However, since the study of modality gap geometry and connection to the cross-modal transferability phenomenon is one of our main contributions, we agree it is useful to provide more information about the modality gap geometry. **We now included distribution plots of the four statistics that reveal important properties of multi-modal embedding space geometry in Appendix A.1.** These distributions provide better evidence than the mean and standard deviation reported in the main paper. For example, we can see that when using CLIP on COCO data, almost all the gaps have a magnitude between 1.13 and 1.23. We hope these added distribution plots provide stronger support for our conclusion: the modality gap can be approximated by a constant vector orthogonal to the image and text embedding spaces.
>
> > The findings (structure of the embeddings) are empirical and not justified by the (contrastive) learning scheme. … About the “constant” modality gap: is there a theoretical justification of it? How general is this phenomenon? Is it a consequence of using contrastive learning?
> >
>
> *How general is the “constant” modality gap phenomenon:*
>
> **In Table 1 and Appendix Figure 4, we show that the “constant” modality gap phenomenon pervasively holds across various contrastive multi-modal models trained on various datasets**, including CLIP (image - text) on COCO and ImageNet, VideoCLIP (video - text), ConVIRT (medical image - text), and CLASP (bio sequence - text). These findings suggest that cross-modal transferability should be a pervasive phenomenon in multi-modal contrastive learning, and it is possible to diagnose models trained on other modalities with language.
>
> *Theoretical justification of the “constant” modality gap:*
>
> Unfortunately, we do not have a theoretical explanation of the constant modality gap geometry. This is a challenging process for two reasons. First, theoretical analysis of the embedding space geometry remains a challenge, even for uni-modality. Moreover, based on the analysis from Liang et al., 2022, the modality gap geometry is caused by the combined effect of model initialization and contrastive learning optimization, not just optimization.
>
> We agree that it would be very interesting to show the theoretical explanation of the constant modality gap, but so far we don’t have it, and this is out of the scope of this work. We clarify that the contribution of this work is to explain cross-modal transferability given the empirical observations of the modality gap geometry, and further propose three novel tasks related to model diagnosis and rectification. We hope our work inspires future work to thoroughly study the multi-modal representation space geometry, given the value we found for downstream applications.
>
> **Nonetheless, we have obtained some additional theoretical insights into the modality gap and cross-modal transferability in Appendix A.4.** This theory connects multi-modal contrastive loss to image-text connection graph partitioning. Specifically, we prove that optimizing a CLIP-like multi-modal contrastive loss is equivalent to minimizing a matrix form loss $\|P-FG^T\|\_F^2$, where P, F, G are image-text connection probabilities, image features, and text features, respectively. Detailed definitions can be found in Appendix A.4. Studying this matrix-form minimization problem provides intuitions about the existence of the modality gap, but cannot justify the “constant” gap. It also provides a complementary understanding of the cross-modal transferability phenomenon. We hope this context will add additional value and provide useful intuition for future works.

---

> > ### Author Response · Authors · 2022-11-10
> > **[2/2] Response to Reviewer Wtjn**
> >
> > **More in-depth analysis of three tasks**
> >
> > > The analysis of the tasks is also not very thorough… ****The three tasks are more illustrations of the multimodal embedding properties than complete studies: they are not fully developed or analyzed. … The results show how the embedding structure can be exploited on three cross-modal tasks. However, I found their analysis a little short: for instance, the literature about feature importance is quite large, even in a multimodal setting [1] and the approach should have been compared to others. [1] Joshi, G., Walambe, R., & Kotecha, K. (2021). A review on explainability in multimodal deep neural nets. IEEE Access, 9.
> > >
> >
> > Thanks for pointing out this paper summarizing many multi-modal interpretation techniques. This work is related so we included it in the paper, but not directly comparable because it does not tackle the same problem. These works focus on instance-level interpretation by finding important features concerning individual examples; we focus on class-level interpretation that aims to identify which attributes generally bias the model predictions to certain classes. Moreover, previous interpretation methods require modifications in model architectures or complex post-processing of model outputs to make the interpretation human-understandable, while our work feeds language into the model and gets meaningful interpretations without pre-processing or post-processing.
> >
> > However, we agree that additional analysis could be helpful, so we added a more comparable baseline, JTT [1], in the model rectification process. Results are shown in **Table 6**. Compared to GDRO, JTT does not require expensive attribute annotation. Our method not only outperforms the baseline but also does not require any visual data.
> >
> > Additionally, we clarify that model diagnosis and rectification is a new area with only a few works. Our work points out a fundamental limitation of previous works about model diagnosis —— the lack of data, and circumvents this challenge by exploiting the unique property of language modality and using natural language to generate test data. Moreover, all the existing works on model diagnosis do not provide a solution to rectify the discovered errors, which is a practical and challenging problem. Our work also proposes a simple solution to this problem by generating additional data using language and continuing to train the model on these text inputs, which works very well and outperforms strong baseline methods.
> >
> > **Finally, we clarify that our contribution is more fundamental than just the three tasks. Our main contribution is theoretically and empirically demonstrating a pervasive phenomenon in multi-modal contrastive learning —— cross-modal transferability. This opens up a new direction of capabilities in using language to diagnose and rectify vision models, which improves in terms of both ease of diagnosis, and diagnosing concepts for which we have limited or no image data. We illustrate examples of how it can be used through the three tasks, but in the scope of this work we cannot explore each of these fully, since each of them can be their own significant works. Our work opens up possibilities for future works to expand on each of these in depth.**
> >
> > We again thank Reviewer Wtjn for their review of our manuscript, which was very helpful in improving the paper. We hope the above responses and changes to our manuscript adequately address your concerns, and that you may be willing to improve your rating as a result. Please let us know if you have further questions or concerns!

---

> ### Author Response · Authors · 2022-11-17
> **We would like to hear back from Reviewer Wtjn**
>
> Dear reviewer Wtjn,
>
> We would like to follow up to see if our response addresses your concerns or if you have any further questions. We would really appreciate the opportunity to discuss this further if our response has not already addressed your concerns. Thank you very much!

---

> ### Comment · Reviewer_Wtjn · 2022-11-28
> **Response to rebuttal**
>
> First I would like to thank the authors for providing motivated and detailed answers to my concerns. Here are a few comments on them.
>
> * **The geometry of the embedding space.** I appreciated the new presentation of the multidimensional gap distribution using histograms that gives more details about its behavior. It reveals however, for me, that the hypothesis that the gap is constant is not that obvious, especially when observing its direction.  The gap distribution seems to follow a rather regular random pattern, but maybe a little bit more variant than a constant: this is why I would have expected a more detailed statistical analysis and modeling, and more rigorous statistical tests to identify a family of models, for instance.
>
> * **Theoretical justification.** I agree that the question is difficult, but if this constant gap phenomenon is shared by many multidimensional embeddings, a natural question is to look for a reason. Its discovery is empirical  and a little bit magical: it limits the reliable exploitation of this property. The new A.4 appendix is a first attempt to address this but clearly not sufficient. This weak theoretical justification is for me a shortcoming of the paper, although the three studied applications show interesting preliminary results. This is why I keep my rating unchanged.

---

> > ### Author Response · Authors · 2022-12-11
> > **Following up for reviewer Wtjn**
> >
> > Dear reviewer Wtjn,
> >
> > Thanks a lot for your reply! Per your suggestion, we now included the chi-square test that further demonstrates the modality gap approximates a constant vector in the multi-modal representation space.
> >
> > Let's denote the gap vector as $g$, and denote $g_i$ as the $i$-th dimension of the vector $g$. For every dimension, we now include a chi-square test that verifies $var[g_i] < \sigma^2$ for a small $\sigma$. The statistical test process is listed in the following:
> >
> > 1. We compute test statistic $T = (N-1) (s/\sigma)^2$, where $s$ is the empirical standard deviation of $g_i$ computed on the dataset, $N$ is the dataset size, and $\sigma$ is a pre-set small target standard deviation.
> > 2. We verify $T$ is smaller than $\chi^2_{1-\alpha, N-1}$, where $\chi^2_{1-\alpha, N-1}$ is critical value of chi-square distribution with significance level $\alpha$ (the probability of test failure). We use $\alpha=0.05$.
> >
> > The statistical test holds if we choose $\sigma^2=0.0044$ for CLIP COCO, $\sigma^2=0.0056$ for CLIP ImageNet, $\sigma^2=0.0021$ for VideoCLIP, $\sigma^2=0.0089$ for ConVIRT, and $\sigma^2=0.015$ for CLASP. These small values demonstrate that every dimension of the gap vector is almost constant, and our finding that the modality gap approximates a constant pervasively holds for various contrastive models trained on different modalities.
> >
> > Meanwhile, we hope to clarify that studying the geometry of any representation learning methods are significant open questions, and while our work is not able to completely solve them, it takes important first steps and provides significant new insights that we believe are valuable to the field and can inspire fruitful future work. We believe these contributions are significant enough to be disseminated alone as a useful publication in the field.
> >
> > We are eager to answer any additional questions.

---

### Official Review · Reviewer_yEKt · 2022-10-24

**Confidence:** 4
**Correctness:** 4
**Technical Novelty And Significance:** 4
**Empirical Novelty And Significance:** 3
**Recommendation:** 6

**Clarity, Quality, Novelty And Reproducibility:**

The ideas in paper are delivered clearly and easy to follow. The proof of theory part for me is enough, clear and correct.

Table 6 is a bit hard to follow since both, error slice and the best accuracy are highlighted in bold.
Also in Table 6, I am not sure if DRO’s performances is taking from original paper or by author’s own implementation. The authors use 3 public datasets (WaterBird, FairFace, dSpitesV), and provides the source code which makes their work reproducible.

**Strength And Weaknesses:**

Strengths

[S1] Ideas are presented with full examples and explanation then it makes paper easy to follow.

[S2] The main contribution of the paper is the theoretical and empirical verification  of the ability to use text as a proxy for image inputs.

[S3] The novelty of the idea to use natural language (which is deemed to be inherently interpretable) to diagnose machine learning models and at the same time addressing the problem of lacking diverge real life vision data.

Weaknesses

[W1] Prior knowledge needed to select a suitable attribute set A.

[W2] Missing comparison to SoA.

Details
- The way of choosing attribute set A initially could be a concern. It seems that the degree of diversity and size of the generated text input set partially depends on the chosen of attribute set A. Without any prior knowledge of spurious correlation within the dataset, it would be hard to choose a really 'related' set A. For example, in the dataset WaterBird, it is widely known the spurious correlation is the bird class and the background, then the attribute was picked are 'bird species’ and ‘places’. In the case of unknown prior knowledge, one might fail to point out the expensive attributes.
- It is hard to judge whether the method has ‘diagnosed’ vision classifiers thoroughly. The authors compare to GDRO which (requires annotations) and  focuses on increasing the worst-group accuracy, which is not addressed by Rectify. Only generating more data on error slice and keep continue training is not expected to help when a model is already biased. There are other works that the authors reference, such as Just Train Twice (Liu et al, 2021), which also do not require annotations.

**Summary Of The Paper:**

Based on observation in Liang el al., 2022 on the modality gap, the paper further proves that the modality gap in multi-modal settings has not influence on the prediction of classifiers, and thus enables cross-modal transferability. The papers further shows that texts can be used as a proxy for image inputs, and provides a method (DrML) to use natural language to identify data subsets (error slices) and the most influential attribute. Their approach is validated on 3 benchmark datasets and shows at some certain level being more convenient than given baselines since this method can diagnose models without requiring diverge visual data collection/creation. The paper further presents a method to rectify undesirable model behaviours based on generating a larger language dataset related to the error slice and continue model training on these.

**Summary Of The Review:**

The paper introduces a novel idea of how to use natural language to diagnose vision classifiers. This method can be understood easily by practitioners, and addresses the lack of out-of-distribution data in one modality. However, there are still some concerns w.r.t. to the method and comparison to SoA.
To the best of my knowledge, this is the first work being able to use natural language to discover error slice of vision classifier and extending language data instead of vision data to correct data bias problem. Other works do research on multi-modal and the phenomenon of modality gap but not emphasize on purpose of diagnosing model. My biggest concern is that author does not provide any proper reasons how they could pick the attribute set. The choice of the attribute set may affect the method performance significantly. In their experiments on the effectiveness of the rectifying method, the paper misses the comparison to related work, they use only one other method (GDRO), while stronger baselines are available.

Edited after author rebuttal:
I appreciate the authors effort on addressing my concerns and their improvements of the paper. Specifically, the authors added a comparison to SoA (experiments) and a discussion on attribute selection (in the appendix). I adjusted my rating. I would appreciate a short note on the manual need for attribute selection in the main body of the paper.

---

> ### Author Response · Authors · 2022-11-10
> **[1/3] Response to Reviewer yEKt**
>
> We thank Reviewer yEKt for reviewing our paper and providing helpful feedback on our work. We address Reviewer yEKt’s concerns below.
>
> **Is using attributes an appropriate and robust choice for model diagnosis?**
>
> > My biggest concern is that author does not provide any proper reasons how they could pick the attribute set. The choice of the attribute set may affect the method performance significantly. … It seems that the degree of diversity and size of the generated text input set partially depends on the chosen of attribute set A. Without any prior knowledge of spurious correlation within the dataset, it would be hard to choose a really 'related' set A.
>
> Thank you for bringing up this question. **While we agree that the practical utility of our method is dependent on how attributes (or more generally, text prompts for diagnosis) are chosen, we clarify that our main contribution is an entirely new way in which text prompts can be used to diagnose vision models. This enables improvements in terms of both ease of diagnosis, and diagnosing concepts for which we have limited or no image data, and the benefits scale with better attribute selection methods that can be independently developed for specific datasets, outside the scope of this work.** Following, we provide additional discussion first on the appropriateness and robustness of attribute selection specifically, and then on its broader context in our work. We have also included this discussion in the revised paper (**Appendix B.6**).
>
> First, we clarify that in our experiments, we did not cherry-pick attributes and just used the known attributes from the data curation process of the three datasets. More broadly, we argue that it is not hard to define a meaningful set of attributes given a specific task; there are many known attributes given any dataset, and sometimes there may be specific attributes of interest for which we wish to test model vulnerability. For example, for a self-driving car application, we can easily come up with attributes such as weather, traffics, pedestrians, buildings, etc. For any class in ImageNet classification, such as guitar, it is straightforward to think about its color, material, location, size, etc. We agree that an initial set of chosen attributes may not be perfect for reflecting all the essential errors, and better attribute selection can reveal more model vulnerabilities. Therefore, this process can be improved with human involvement to iteratively design better attributes based on the model feedback, which can be useful for future works.
>
> While attribute selection is important and may require human involvement, our method is still very useful because we provide an easy way to test the model under many cases. Like software testing, there is generally no free lunch for model diagnosis, and it is impossible to design a general diagnosis framework for any task without any prior knowledge. We have already significantly reduced the diagnosis cost compared to previous works. Previous works all assume a large collection of labeled images is available for model testing, which is unrealistic given the extreme difficulty of collecting diverse image inputs and the cost of data annotation. Our method instead provides a way to test sensitive attributes for which you may even have no image data. For any specific task, it is always much easier to come up with a meaningful set of attributes and then generate a large collection of novel text inputs by combining different attributes than collecting corresponding images, thanks to the easy-to-manipulate and compositional nature of the language modality. More importantly, the combination of defined attributes naturally defines human-interpretable data slices, whereas image-based slice discovery methods do not directly provide a text summary of the error slice.
>
> Finally, we hope to clarify that one of the main contributions of our work is to theoretically and empirically demonstrate a pervasive phenomenon in multi-modal contrastive learning —— cross-modal transferability, which allows texts to be effective proxies for images. Our method performance, in terms of correlation strength between model performances on images and corresponding texts, is independent of attribute selection. It is just more errors can be discovered with more human involvement in this process. Moreover, it is possible to collect a large set of text inputs in a different way to diagnose vision models instead of using the attribute-based combination. For example, one may be able to prompt large language models such as GPT-3 in a few-shot fashion to generate a large set of descriptions of certain classes, and then feed these inputs into vision models for diagnosis. We leave this to future work. Overall, diagnosing vision models using text modality is much more desirable than image modality, because language enables us to easily generate realistic and diverse inputs with better control and manipulation.

---

> > ### Author Response · Authors · 2022-11-10
> > **[2/3] Response to Reviewer yEKt**
> >
> > **More details about rectifying model misbehaviors**
> >
> > > I am not sure if DRO’s performances is taking from original paper or by author’s own implementation.
> > >
> >
> > **We adopt the official GDRO loss implementation to our code base using all the same hyperparameters they reported in the paper**. We updated the implementation details, such as hyperparameters, in **Appendix C.3** of the revised manuscript. Implementation details can also be found at [https://anonymous.4open.science/r/diagnosis/src/utils/dro_loss.py](https://anonymous.4open.science/r/diagnosis/src/utils/dro_loss.py).
> >
> > > In their experiments on the effectiveness of the rectifying method, the paper misses the comparison to related work, they use only one other method (GDRO), while stronger baselines are available… ****There are other works that the authors reference, such as Just Train Twice (Liu et al, 2021), which also do not require annotations.
> > >
> >
> > Thanks for pointing out this baseline. We now included this baseline, JTT (Just Train Twice), in the revised paper. **From Table 6 in the revised main text, we can clearly see that our method outperforms this baseline.** We also provided more implementation details of JTT in **Appendix C.3.**
> >
> > > Only generating more data on error slice and keep continue training is not expected to help when a model is already biased.
> > >
> >
> > We first provide more details about the model rectification process to make it easier to be understood:
> >
> > In addition to discovering errors during model diagnosis, how to rectify these errors is a practical but challenging problem, which is seldomly addressed in existing works about the model diagnosis. Our finding about cross-modal transferability enables us to rectify undesirable behaviors of vision classifiers through language. Here we propose a simple solution where we generate additional data that the model fails using language and continue training the model on these synthesized data.
> >
> > Given the error slices  $\mathbb{S} = \{\mathcal{S} \subseteq \mathcal{X}| e(\mathcal{S}) \gg e(\mathcal{X}) \}$ discovered, we aim to rectify model performance on these error slices by minimizing $|\mathbb{S}|$. For each $\mathcal{S} \in \mathbb{S}$ defined by a list of attributes, we generate a large set of natural language inputs related to this slice $\mathcal{Y}\_{\mathcal{S}}$ through attribute composition and prompt manipulation (Appendix B) and continue training the model on these text inputs $\mathcal{Y}\_{\mathcal{S}}$. We continue training the model using the same hyperparameters as if the model is trained on corresponding images, since we have proved that texts are effective proxies of images. This simple strategy significantly improves model performances on corresponding image error slices with minimal impact on other data, and has a distinct advantage that no visual data is required for rectification.
> >
> > With the process described above, we demonstrate that our method indeed successfully rectifies the biased model under three typical settings: spurious correlation, low-data drift, and unseen data. **The reason why generating more data on error slices and continuing training work so well, despite a biased model, is that texts are good proxies for images, given our theoretical and empirical explanations about cross-modal transferability.** We agree that maybe not all the errors can be rectified by correcting data bias through continuing training, but we believe that our method addresses an important challenge in a wide range of cases. Moreover, we are the first work proposing an easy approach to mitigate the errors after diagnosis, even without requiring any visual data in this process, while previous works about model diagnosis do not provide a solution for rectification, and previous works about robust training cannot fix errors on the unseen data.

---

> > > ### Author Response · Authors · 2022-11-10
> > > **[3/3] Response to Reviewer yEKt**
> > >
> > > > It is hard to judge whether the method has ‘diagnosed’ vision classifiers thoroughly. The authors compare to GDRO which (requires annotations) and focuses on increasing the worst-group accuracy, which is not addressed by Rectify.
> > > >
> > >
> > > Thank you for the question. We first clarify that GDRO is not a directly comparable baseline because it requires annotations. Per your suggestion, we now included a more comparable baseline, JTT, and show that our method significantly outperforms this baseline in **Table 6**. We also highlight another distinct advantage of our method: we can rectify models when no visual data is available, while JTT and GDRO baseline cannot. Also, when group information is available, our method is complementary to GDRO and JTT, because GDRO and JTT focus on improving optimization, and we focus on improving data. We can do both simultaneously to get an even more robust model.
> > >
> > > We agree that it is hard to judge whether our method has found all the errors in the model, because we do not know all the ground-truth errors, and it is impossible to enumerate all the test cases. As a result, we may not fix all the errors during the rectifying process. Nonetheless, as we show, our rectifying method has minimal impact on the well-performed data subgroups, and the model keeps improving during the rectification process as we improve the training data. When there is more human involvement in the diagnosis process by selecting better attributes and designing better prompts, this can help us more comprehensively discover errors and get much more robust models by rectifying them.
> > >
> > > We again thank Reviewer yEKt for their review of our manuscript, which was very helpful in improving the paper. We hope the above responses and changes to our manuscript adequately address your concerns, and that you may be willing to improve your rating as a result. Please let us know if you have further questions or concerns!

---

> ### Author Response · Authors · 2022-11-17
> **We would like to hear back from Reviewer yEKt**
>
> Dear reviewer yEKt,
>
> We would like to follow up to see if our response addresses your concerns or if you have any further questions. We would really appreciate the opportunity to discuss this further if our response has not already addressed your concerns. Thank you very much!

---

> ### Author Response · Authors · 2022-12-11
> **Following up for reviewer yEKt**
>
> Dear reviewer yEKt, we are grateful for your suggestions, and hope our response and updated PDF were helpful in addressing the concerns. We are eager to answer any additional questions.

---

### Official Review · Reviewer_LGzN · 2022-10-29

**Confidence:** 3
**Correctness:** 3
**Technical Novelty And Significance:** 3
**Empirical Novelty And Significance:** 3
**Recommendation:** 6

**Clarity, Quality, Novelty And Reproducibility:**

Clarity : I could clearly understand, except for a few parts.
- I don’ t know the exact meaning of the term “closing the modality gap”. I wonder if it simply means reducing the modality gap or includes other meanings.
- I wonder what the process of figuring out influential attributes means in your framework, DrML. As far as I understand, it seems that the model is rectified using error slices but I don't know how the influential attribute is used. Is this just to provide additional information to humans?

Quality : The main claims are well-supported by various experiments and theoretical proof.

Novelty : This paper looks to contribute some new ideas.

Reproducibility : Most of required information for reproducibility is provided.


**Strength And Weaknesses:**

**Strength**
- They prove that the phenomenon of pervasive cross-modal transferability occurs when the modality gap meets certain conditions by empirically and theoretically.
- It was made human-interpretable by explaining the model's error using languages as well as images.
- The proposed framework further rectifies the behavior of the models.

**Weakness**
- The explanation of the process of correcting the misbehavior of the model using language may need more detail.
- Few parts of the proposed framework are not justified well (Details are in Clarity 2.)



**Summary Of The Paper:**

This paper explains why cross-modal transferability happens even if there is a modality gap. Authors have shown this empirically on various datasets and have also theoretically proved it as well. By proving this, they demonstrate that text can be a good proxy for images. Based on this assumption, they introduce a new framework, DrML, which can discover error slices, identify influential attributes and correct misbehavior of the model.



**Summary Of The Review:**

This paper demonstrates their assumptions about cross-modal transferability in several ways, and experiments are also conducted on several datasets. Also, they introduce a new framework, DrML, that can rectify the model’s misbehavior. In sum, I think the proposed idea looks novel and may shed new light on diagnosing models. However, there are some questionable points when understanding this paper. I think it could be accepted if these are improved.

---

> ### Author Response · Authors · 2022-11-10
> **[1/2] Response to Reviewer LGzN**
>
> We thank Reviewer LGzN for their positive comments and for providing thoughtful feedback on our work. We address Reviewer LGzN’s concerns below.
>
> **The meaning of closing the modality gap**
>
> > I don’t know the exact meaning of the term “closing the modality gap”. I wonder if it simply means reducing the modality gap or includes other meanings.
> >
>
> Yes, it means reducing the modality gap. The simple intuition is that since we observe that the modality gap approximates a constant vector, we can remove this constant vector to make embeddings from two modalities the same and not able to be differentiated by the classifier. We describe it formally in the **last part of Section 2.1**. Here we provide more explanations to make it easier to be understood. Denote $x$ and $y$ are embeddings from two modalities. During training, instead of feeding $x$ to the classifier, we feed it with $x’ = x − E_x[x]$; during the cross-modal evaluation, we feed $y’ = y − E_y[y]$ instead of y. In that way, the inconsistency during cross-modal evaluation caused by the modality gap is removed, because $x’ - y’ = x − E_x[x]- y − E_y[y] = (x - y) - (E_x [x] - E_y [y]) = g - g = 0$, so $x’ = y’$. With this strategy, we observe additional improvements in cross-modal transferability compared to training with the gap.
>
> We have added clarification of the term “closing the modality gap” to the paper.
>
> **The use case of figuring out influential attributes**
>
> > I wonder what the process of figuring out influential attributes means in your framework, DrML. As far as I understand, it seems that the model is rectified using error slices but I don't know how the influential attribute is used. Is this just to provide additional information to humans?
> >
>
> Figuring out influential attributes indeed provides additional information to humans, but also helps us discover error slices, because the attribute by itself provides information about the space of potential error slices. For example, when figuring out that the attribute “ocean” significantly biases the model towards “waterbird”, we would naturally think about error slices could be “landbirds in ocean”, “only an ocean background without any birds”, or “landbirds in a background similar to the ocean such as a lake or raining forest”.
>
> On the other hand, if we discover error slices such as “waterbird in land” and “landbird in water”, it can inspire us to think that the background causes these errors. Then we can use our framework to compute attribute influences of different backgrounds and thereby explain why the model fails on these slices.
>
> In summary, error slice discovery finds specific subgroups where a model fails, while attributes provide high-level explanations of the errors. To reliably deploy a model in the wild, understanding when and why the model fails are both critical. Therefore, discovering error slices and identifying influential attributes are very important **complementary** applications, with the same ultimate goal —— diagnosing the model.
>
> We have revised our manuscript and included this discussion now.

---

> > ### Author Response · Authors · 2022-11-10
> > **[2/2] Response to Reviewer LGzN**
> >
> > **More details about rectifying model misbehaviors**
> >
> > > The explanation of the process of correcting the misbehavior of the model using language may need more detail.
> > >
> >
> > Thank you for your suggestion. We have revised the paper and included more details in the rectification process:
> >
> > In addition to discovering errors during model diagnosis, how to rectify these errors is a practical but challenging problem, which is seldomly addressed in existing works about the model diagnosis. Our finding about cross-modal transferability enables us to rectify undesirable behaviors of vision classifiers through language. Here we propose a simple solution where we generate additional data that the model fails using language and continue training the model on these synthesized data.
> >
> > Given the error slices  $\mathbb{S} = \{\mathcal{S} \subseteq \mathcal{X}| e(\mathcal{S}) \gg e(\mathcal{X}) \}$ discovered, we aim to rectify model performance on these error slices by minimizing $|\mathbb{S}|$. For each $\mathcal{S} \in \mathbb{S}$ defined by a list of attributes, we generate a large set of natural language inputs related to this slice $\mathcal{Y}\_{\mathcal{S}}$ through attribute composition and prompt manipulation (Appendix B) and continue training the model on these text inputs $\mathcal{Y}\_{\mathcal{S}}$. We continue training the model using the same hyperparameters as if the model is trained on corresponding images, since we have proved that texts are effective proxies of images. This simple strategy significantly improves model performances on corresponding image error slices with minimal impact on other data, and has a distinct advantage that no visual data is required for rectification.
> >
> > Moreover, we provided implementation details of our rectification process in **Appendix B.4**. Per reviewer yEKt’s suggestion, we also included a more comparable rectification baseline, JTT [1], which does not require expensive attribute annotation for images during training compared to GDRO. Results are updated in **Table 6** in the main text, and our method outperforms this baseline. The implementation details of baseline methods (GDRO and JTT) are shown in **Appendix C.3**.
> >
> > Thank you again for your feedback, which was very helpful in improving the paper. We hope the above responses and changes to our manuscript adequately address your concerns, and that you may be willing to improve your rating as a result. Please let us know if you have further questions or concerns!
> >
> > **References:**
> >
> > [1] Liu, Evan Z., et al. "Just train twice: Improving group robustness without training group information." ICML 2021.

---

> ### Author Response · Authors · 2022-11-17
> **We would like to hear back from Reviewer LGzN**
>
> Dear reviewer LGzN,
>
> We would like to follow up to see if our response addresses your concerns or if you have any further questions. We would really appreciate the opportunity to discuss this further if our response has not already addressed your concerns. Thank you very much!

---

> ### Author Response · Authors · 2022-12-11
> **Following up for reviewer LGzN**
>
> Dear reviewer LGzN, we are grateful for your suggestions, and hope our response and updated PDF were helpful in addressing the concerns. We are eager to answer any additional questions.

---

### Author Response · Authors · 2022-11-10
**General Response**

We thank the reviewers for their thoughtful and constructive review of our manuscript. We were encouraged to hear that all the reviewers found the task and method we propose that uses language to diagnose vision models is novel, original, and valuable (LGzN, yEKt, Wtjn), that they view our experiments and analyses as insightful and well-justified (LGzN, yEKt, Wtjn), and that all reviewers found our paper well-written and easy-to-follow (LGzN, yEKt, Wtjn).

**As pointed out by multiple reviewers, we would like to highlight that the main contribution of our work to the deep learning and vision-language community is that we theoretically and empirically demonstrate a pervasive phenomenon in multi-modal contrastive learning —— cross-modal transferability. This opens up a new direction of capabilities in using language to diagnose and rectify vision models, which improves in terms of both ease of diagnosis, and diagnosing concepts for which we have limited or no image data.** We illustrate examples of how it can be used through the three tasks, but in the scope of this work we cannot explore each of these fully, and our work opens up possibilities for future works to expand on each of these in depth. Our work also has a broader impact on all the works built on multi-modal contrastive representation space such as DALLE-2 and provides a unique contribution towards achieving responsible and trustworthy vision systems.

In response to feedback, we provide individual responses below to each reviewer, and **we carefully updated the paper based on the reviewers’ suggestions (updates highlighted in blue)**. We would again like to thank all the reviewers for their time and feedback, and we hope that our responses and the revised manuscript adequately address all the concerns. Please let us know if you have further questions or concerns!

---

### Decision · Program_Chairs · 2023-01-20

**Decision:**

Accept: poster

**Justification For Why Not Higher Score:**

The main reason to accept the paper as a poster and not as a spotlight is the moderate novelty.

**Justification For Why Not Lower Score:**

We consider the ideas and experiments of this paper are interesting for the ICLR community.

**Metareview: Summary, Strengths And Weaknesses:**

The paper addresses the problem of diagnosing vision classifiers using natural language. The paper presents a theoretical formulation, a method for vision model diagnosis, and experiments on diagnosing models and rectifying undesirable model biases. In terms of reproducibility, the code is provided by an anonymous repository.

The reviewers valued the theoretical and empirical contribution of the work, and the overall clarity of the paper (a few unclear aspects on the first submission were pointed out by some of the reviewers; authors added further explanations in the revised paper accordingly). In terms of weaknesses, the reviewers pointed out four main issues: (1) the arbitrarity on the selection of the attribute set A; (2) the requirement of more comparisons with SoA (a specific suggestion of method to compare with was made by one of the reviewers); (3) the need for a more solid justification of the findings (in particular, a connection between the empirical results and the learning scheme); (4) the need for a deeper analysis of the obtained results. The authors addressed the different points in their feedback and also made some changes in the paper.

The paper was discussed in a virtual meeting. Overall, we agreed that the contributions of the paper are valuable for the paper to be accepted at ICLR. Authors should double check all the recommendations made by the reviewers and make sure the final version of the paper includes all the reviewers' suggestions.


**Note From Pc:**

if the above contains the word "oral" or "spotlight" please see: "oral" presentation means -> notable-top-5% and "spotlight" means -> notable-top-25%. As stated in our emails, we are disassociating presentation type from AC recommendations

**Summary Of Ac-Reviewer Meeting:**

The impression of the reviewers was that the paper is an interesting work, but there were some concerns around the novelty. In particular reviewers pointed out the fact that some of the theoretical findings were somehow straightforward and the need for a deeper understanding of the results. After checking the authors feedback and the new version of the paper the majority of the reviewers were positive about accepting the paper.